# DATA EFFICACY FOR LANGUAGE MODEL TRAINING

## ABSTRACT

Data is fundamental to the training of language models (LM). Recent research has focused on data efficiency, aiming to reduce data scale without compromising model performance. However, **data efficacy**, emphasizes improving model performance by optimizing the utilization of training data, is an area that remains underexplored. To enhance it, we propose novel methods for both data ordering and data scoring. For data ordering, we design *Folding Ordering (FO)*, which addresses challenges such as data distribution bias and model forgetting introduced by traditional curriculum learning. For data scoring, we present *Learnability-Quality Scoring (LQS)*, the first method specifically designed to support both data ordering and selection. To further establish the foundation for data efficacy, a general paradigm, **DELT** (**D**ata **E**fficacy for **L**M **T**raining), is introduced to underscore the importance of training data utilization. It comprises two essential modules: data scoring and data ordering, along with one optional module of data selection. This primarily enables DELT to improve data efficacy as well as efficiency. Comprehensive experiments validate our approach, demonstrating that FO and LQS significantly improve LM performance across various settings, consistently surpassing existing baselines. We believe that data efficacy, which aims to fully harness data value without altering data scale and model size to benefit model performance, is a promising foundational area in LM training.

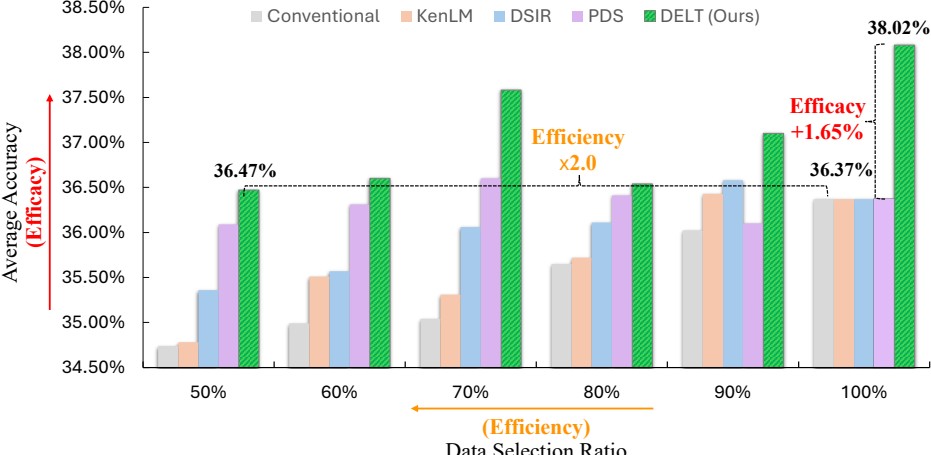

Figure 1: Average result across 8 benchmarks for different methods. High performance at the same selection ratio indicates high efficacy, while achieving similar performance with a smaller selection ratio demonstrates high efficiency. Our method excels in both efficacy and efficiency.

## 1    INTRODUCTION

The significance of language models (Ouyang et al., 2022; Achiam et al., 2023; Dubey et al., 2024) is immense in modern computational applications. From natural language processing tasks such as translation (Hirschberg & Manning, 2015) and sentiment analysis (Gunasekaran, 2023) to more complex applications like automated reasoning (Yu et al., 2024a) and AI agents (Kusal et al., 2022), language models have revolutionized the way machines understand, generate, and interact with human beings using natural language. To empower language models having these abilities, data is central to their training and serves as the foundation from which models learn knowledge based

on linguistic patterns and structures. Consequently, meticulous data curation is essential to ensure consistently high model performance across various applications.

In data curation, we focus on **data efficacy**, defined as *optimizing the utilization of training data to improve model performance*. This direction stands alongside **data efficiency**, which aims to *reduce the scale of training data without sacrificing model performance* (Albalak et al., 2024; Xie et al., 2023; Gu et al., 2025). In standard data efficiency pipelines, once a subset is selected, the retained samples are typically treated uniformly and presented to the model in random order. In contrast, data efficacy explores how to exploit those samples more effectively (e.g., via informed ordering, weighting, or scheduling). Although still nascent, its promise is illustrated by curriculum learning (Wang et al., 2021), which presents examples from easy to hard.

In this context, we observe that the latest generation of language models (OpenAI; Dubey et al., 2024; Team et al., 2023) are typically trained for only a few epochs, often just one, due to the immense scale of their training datasets and the constraints of available computing power. Unlike earlier models (Cho et al., 2014; Hochreiter & Schmidhuber, 1997), which were trained over many epochs and often suffered from overfitting, these newer models adhere to the principles of the scaling law (Kaplan et al., 2020). This shift is further supported by the findings of QQT (Goyal et al., 2024), which demonstrate that the utility of high-quality data diminishes rapidly when reused extensively. In other words, it is more beneficial to train on a vast amount of data over a few epochs than to rely on repeated use of high-quality data across many epochs. Therefore, *the optimal utilization of training datasets becomes crucial for maximizing the performance of language models trained within such limited epochs.*

Expanding on these insights, we propose novel methods for data ordering and scoring to enhance data efficacy. For *data ordering*, we introduce the **Folding Ordering (FO)** method, a new data ordering approach inspired by human learning practices, where similar content is revisited multiple times. Unlike traditional Curriculum-Learning-based methods (Campos, 2021; Wang et al., 2021) that directly sort data and risk inner distribution biases and model forgetting in limited epochs, FO significantly mitigates these challenges, especially in scenarios involving one or few training epochs for language model training. For *data scoring*, we propose the **Learnability-Quality Score (LQS)**, a method that first integrates data ordering and selection into a unified metric. LQS captures sample learnability through dynamic gradient magnitude changes and evaluates sample quality using static gradient angle information. By combining these two aspects, LQS enables a reasonably optimized organization of samples, which enhances both data efficacy and efficiency. Building on this foundation, we also propose a general paradigm for enhancing **d**ata **e**fficacy for **LM** **t**raining (**DELT**). This paradigm aims to improve model performance by fully harnessing data value without modifying the dataset content or model architecture, making it an almost cost-free approach. Specifically, DELT currently incorporates data scoring and ordering, and it can be seamlessly integrated with data selection to further improve data efficiency.

To validate the capability of the introduced paradigm, we incorporate several baseline methods into it, along with the newly designed methods, LQS and FO, for data scoring and ordering, respectively. The key results from Figure 1 highlight that the proposed DELT significantly improves data efficacy in LM training on a set of typical benchmarks. Meanwhile, it outperforms existing methods (Gu et al., 2025; Heafield, 2011) in data efficiency that further boosts LM performance across all selection ratios. The main contributions are as below:

- We introduce an innovative method for data ordering, named Folding Ordering (FO), which optimizes the utilization of training data and mitigates the issues of data distribution bias and model forgetting brought by previous methods.

- We propose a novel method for data scoring, called Learnability-Quality Scoring (LQS), which is the first to consider both data efficacy and efficiency by evaluating each sample based on its learnability and quality.

- We identify the untapped potential of data efficacy in language model training and present a general paradigm DELT. Through comprehensive experiments on mainstream benchmarks, we validate DELT using various data scoring and ordering methods. All implementations of DELT improved performance, with our designs FO and LQS achieving the best results.

Through these contributions, we aim to provide a general paradigm for understanding and applying data efficacy in LM training, paving the way for more efficacious model development practices.

## 2 RELATED WORK

### 2.1 DATA SOURCES

Data source of LM training (Anthropic; Yang, 2007; Dubey et al., 2024; Ouyang et al., 2022; Team et al., 2023) can primarily be categorized into five types: internet data (Rana, 2010), books (Hart, 2004), synthetic data (Nikolenko et al., 2021), physical sensors (Kabadayi et al., 2006), and human perception of the real world. Internet data is the primary source for language model training due to its vast scale. Books and synthetic data offer high quality, but are limited in scale. Data from physical sensors and human perception are in other modalities or still under development. Several studies focus on extracting high-quality datasets for LM training, like C4 (Raffel et al., 2020), RefinedWeb (Penedo et al., 2023), RedPajama (Weber et al., 2024), and RedStone (Chang et al., 2024). All of them utilize an identical data source, CommonCrawl (Rana, 2010), which captures snapshots of web pages from the entire internet at different periods and contains over 200 billion samples to date.

### 2.2 DATA EFFICIENCY

Data efficiency (Heafield, 2011; Gu et al., 2025; Xie et al., 2023; Albalak et al., 2024) focuses on selecting the most relevant data points for inclusion in training dataset and optimizing the performance of the language model. This area includes well-researched strategies such as data selection (Heafield, 2011; Gu et al., 2025; Yu et al., 2024b; Dai et al.), sampling (Xie et al., 2023), denoising (Zhao et al., 2021; Hu et al., 2021), and deduplication (Abbas et al., 2023; Tirumala et al., 2023), all of which aim to select optimal data for efficient model training. The KenLM (Heafield, 2011) trains a fast and small model for perplexity estimation and treats the perplexity as the data difficulty for LM. The PDS (Gu et al., 2025) evaluates the quality of data samples by measuring the consistency of each sample's gradient direction with a reference direction. The DSIR (Xie et al., 2023) develops an importance weight estimator to select a subset of raw data that mirrors the distribution of the target in a specific feature space. The MATES (Yu et al., 2024b) presents a data influence model that continuously adapts to the evolving data preferences of the pre-trained model, selecting the most effective data for the current stage of pre-training. The SemDeDup (Abbas et al., 2023) utilizes embeddings from pre-trained models to identify and remove data pairs that are semantically similar but not exactly identical. All these methods develop strategies to decide whether a sample should be retained or discarded. However, for retained samples, language models train on them equally, without considering differences in criteria.

### 2.3 DATA EFFICACY

Data efficacy, distinct from data efficiency, aims to maximize the performance of language models by optimizing the organization of training data. Curriculum learning, as described by Campos (2021), involves starting with simpler examples and progressively tackling more complex ones, aiding in smoother model convergence. Within curriculum learning, Kim & Lee (2024) presents an attention score to determine the prompt difficulty, and Chang et al. (2021) introduces a soft edit distance to measure sample difficulty. Similarly, annealing learning, as outlined by Dubey et al. (2024), seeks to improve model performance by initially training on a large, noisy dataset and concluding with a small, high-quality dataset. All these methods sort training data directly by difficulty or quality. However, since limited research on data efficacy, there is no established paradigm for effectively organizing training data.

To conclude, data is essential for training language models, and numerous large-scale data sources originate from the internet. Nevertheless, obtaining incremental public data has become challenging due to the slow growth of CommonCrawl (Rana, 2010) snapshots and the increasing presence of AI-generated content online. As language models scale up, effectively leveraging existing data sources becomes vital, which makes data efficacy increasingly important. Despite this, few studies focus on data efficacy in language model training. To address this gap, we propose a general paradigm for it, where curriculum learning and annealing learning are two specific instances.

# 3 METHOD

## 3.1 DATA ORDERING

Existing methods usually utilize the random shuffling method to organize training data. Alternatively, some approaches adopt curriculum learning (Campos, 2021; Kim & Lee, 2024), a sorting-based method inspired by the human learning process, where training begins with simpler samples and gradually progresses to more complex ones. Sorting-based methods can improve training efficiency; however, they often face challenges such as data distribution bias and model forgetting, as analyzed in the Appendix 15, which can ultimately degrade performance.

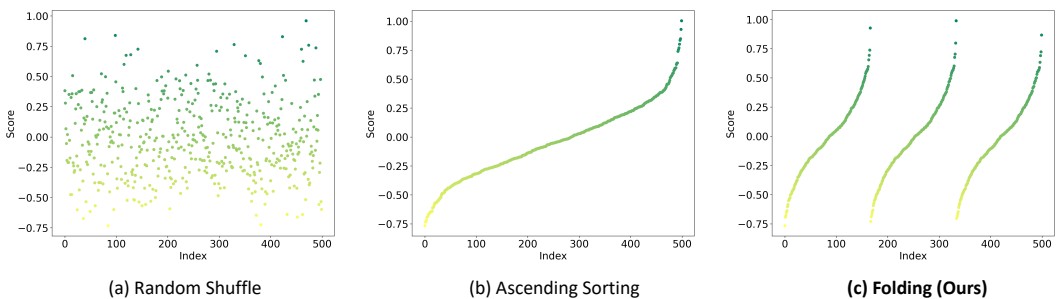

(a) Random Shuffle  (b) Ascending Sorting  (c) **Folding (Ours)**

Figure 2: Illustration of ordering methods. The right one is Folding method, an advanced multi-fold version of the Sorting method. These results are based on 500 random samples from RedPajama.

**Folding** method is proposed to improve training data efficacy and address the negative influences brought by the sorting method. The new method, named ***folding learning***, reorganizes the dataset by repeating the curriculum learning multiple times without duplication. The repeated times is defined as the folding layers $L$. As demonstrated in Figure 2, the folding method samples sorted data $L$ times without replacement at a fixed interval $L$. The permutation function $\pi_{fold}$ is defined in Equation 1, while $\pi_{sort}$ is described in Equation 12. Folding learning not only inherits the benefits from curriculum learning but also mitigates issues of model forgetting, and data distribution bias.

$$\pi_{fold}(\boldsymbol{\gamma}; L) = \bigcup_{\ell=0}^{L-1} \langle \pi_{sort}(\boldsymbol{\gamma})_i \mid i \in \{j \mid j \equiv \ell \pmod{L}, 1 \leq j \leq |\mathcal{D}|\} \rangle \qquad (1)$$

## 3.2 DATA SCORING

Existing methods typically focus on attributes such as quality (Gu et al., 2025), difficulty (Heafield, 2011), noisiness (Zhao et al., 2021; Hu et al., 2021), or diversity (Abbas et al., 2023; Tirumala et al., 2023) to compute scores for data selection. However, these methods, designed primarily for data selection, often focus solely on how *good* a sample is, while overlooking the question of *where* a sample contributes most effectively within the context of the entire dataset.

**Learnability-Quality Scoring (LQS)** is introduced to address this limitation and make the scorer more attentive to the utility of each data sample. By incorporating both *learnability* and *quality*, LQS is not only sensitive to low-quality samples but also better weights the impact of samples during model training. Our method dynamically evaluates how each sample contributes to reducing the downstream loss $J(\boldsymbol{\theta})$ by considering its behavior at different training stages, where $\boldsymbol{\theta} \in \mathbb{R}^N$ represents the parameters of the LM and the $N$ is the dimension of the parameter space.

The ***learnability*** of each data sample represents the difficulty change during model training, as illustrated in Figure 3a. For each training step $t$ from 1 to $T$ on the dataset $\mathcal{D} = \{x_n\}_{n=1}^{|\mathcal{D}|}$, the learnability of a sample $x_n$ (the $n$-th sample in $\mathcal{D}$) is defined as its contribution to reducing the loss over time during training. The learnability is represented as:

$$\mathcal{L}(x_n) = \sum_{t=1}^{T-1} \frac{l_{n,t}}{l_{n,t+1}} = \sum_{t=1}^{T-1} \frac{\|\nabla\ell(x_n, \boldsymbol{\theta}_t)\|}{\|\nabla\ell(x_n, \boldsymbol{\theta}_{t+1})\|}, \qquad (2)$$

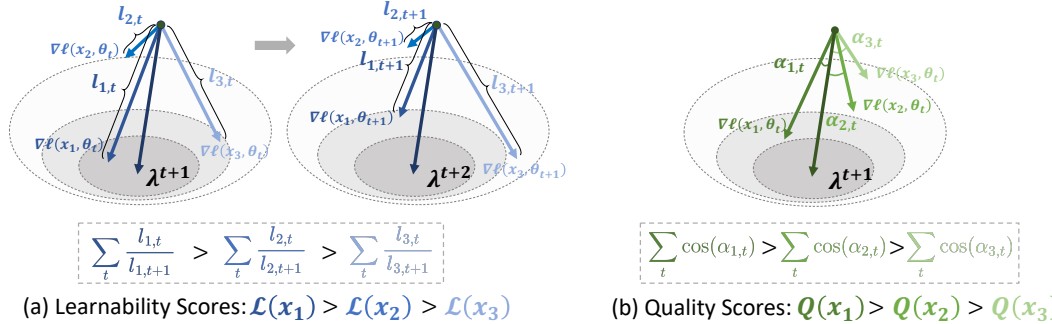

$$\sum_t \frac{l_{1,t}}{l_{1,t+1}} > \sum_t \frac{l_{2,t}}{l_{2,t+1}} > \sum_t \frac{l_{3,t}}{l_{3,t+1}} \qquad \sum_t \cos(\alpha_{1,t}) > \sum_t \cos(\alpha_{2,t}) > \sum_t \cos(\alpha_{3,t})$$

(a) Learnability Scores: $\mathcal{L}(x_1) > \mathcal{L}(x_2) > \mathcal{L}(x_3)$     (b) Quality Scores: $Q(x_1) > Q(x_2) > Q(x_3)$

Figure 3: Illustration of LQS scoring method. The left part demonstrates the calculation of the learnability score, and the right part depicts the computation of the quality score.

where $\nabla\ell(x_n, \boldsymbol{\theta}_t)$ denotes the gradient of loss function for sample $x_n$ at training step $t$, with model parameters $\boldsymbol{\theta}_t$, and $l_{n,t}$ is its magnitude. A high learnability score indicates that the sample significantly reduces the loss in training, particularly if its gradient magnitude is initially high and decreases substantially over time. Such samples are challenging yet beneficial for training, making them more suitable for later stages of training. Conversely, noisy samples or those with unstable gradients yield a low learnability score, enabling their identification and efficient filtering during data selection.

The **quality** of each data sample contributes to data efficacy during model training, as depicted in Figure 3b. It is measured by the consistency of $\nabla\ell(x_n, \boldsymbol{\theta}_t)$ with a target vector $\boldsymbol{\lambda}_{t+1}$ in Equation 4, which represents the average gradient of the loss function for all data at training step $t+1$. The quality score is computed as:

$$Q(x_n) = \sum_{t=1}^{T-1} \cos(\alpha_{n,t}) = \sum_{t=1}^{T-1} \frac{\boldsymbol{\lambda}_{t+1}^\top \nabla\ell(x_n, \boldsymbol{\theta}_t)}{\|\boldsymbol{\lambda}_{t+1}\| \cdot \|\nabla\ell(x_n, \boldsymbol{\theta}_t)\|}, \tag{3}$$

where $\alpha_{n,t}$ is the angle between two vectors. A higher cosine similarity $\cos(\alpha_{n,t})$ indicates that the gradient convergence direction on $x_n$ is more aligned with the target objective $\boldsymbol{\lambda}_{t+1}$, implying a stronger contribution to reducing the loss $J(\boldsymbol{\theta})$. Following Gu et al. (2025), the target vector $\boldsymbol{\lambda}_t$ is:

$$\boldsymbol{\lambda}_t = \begin{cases} \boldsymbol{\lambda}_{t+1} + \nabla J(\boldsymbol{\theta}_t) - \eta \cdot \nabla^2 L(\boldsymbol{\theta}_t, \boldsymbol{\gamma}) \cdot \boldsymbol{\lambda}_{t+1}, & \text{if } t < T \\ \nabla J(\boldsymbol{\theta}_t), & \text{if } t = T \end{cases} \tag{4}$$

Finally, we combine *learnability* and *quality* into a unified function to score data samples. We then provide a more detailed explanation of the Learnability-Quality Scoring (LQS). The Learnability Score $\mathcal{L}(x_n)$ (Equation 2) and the Quality Score $Q(x_n)$ (Equation 3) are proposed, both of which are directly related to the training sample $x_n$ and are used as metrics to evaluate $x_n$.

However, the reliability of these scores is directly influenced by the capability of the model checkpoint. Models with stronger capabilities produce more reliable scores. This capability is linked to the target vector, which is calculated as an average across all samples. When the structural parameters of the model remain unchanged, a stronger model shows greater consistency in the gradient directions of all samples concerning the model parameters, resulting in a larger $\nabla J(\boldsymbol{\theta}_t)$ (Equation 4). This corresponds to a greater magnitude of the target vector. Therefore, we use the magnitude of the target vector to directly measure the model's capability, referred to as the reliability score $R(\boldsymbol{\theta}_{t+1}) = \|\boldsymbol{\lambda}_{t+1}\|$. Finally, we anticipate that a stronger model will assign more weight to the scores. LQS is expressed as a combination of the sample's learnability-quality score and the model's capabilities.

$$\gamma_n = R(\boldsymbol{\theta}_{t+1}) \cdot Q(x_n) \cdot \mathcal{L}(x_n) \tag{5}$$

$$= \sum_{t=1}^{T-1} \|\boldsymbol{\lambda}_{t+1}\| \cdot \frac{\boldsymbol{\lambda}_{t+1}^\top \nabla\ell(x_n, \boldsymbol{\theta}_t)}{\|\boldsymbol{\lambda}_{t+1}\| \cdot \|\nabla\ell(x_n, \boldsymbol{\theta}_t)\|} \cdot \frac{\|\nabla\ell(x_n, \boldsymbol{\theta}_t)\|}{\|\nabla\ell(x_n, \boldsymbol{\theta}_{t+1})\|}$$

$$= \sum_{t=1}^{T-1} \frac{\boldsymbol{\lambda}_{t+1}^\top \nabla\ell(x_n, \boldsymbol{\theta}_t)}{\|\nabla\ell(x_n, \boldsymbol{\theta}_{t+1})\|} \tag{6}$$

For a detailed explanation of the formula, the score vector $\boldsymbol{\gamma}$ is defined as:

$$\boldsymbol{\gamma} = \left\{ \gamma_n \middle| \gamma_n = \sum_{t=1}^{T-1} \frac{\boldsymbol{\lambda}_{t+1}^\top \nabla \ell(x_n, \boldsymbol{\theta}_t)}{\|\nabla \ell(x_n, \boldsymbol{\theta}_{t+1})\|}, 1 \leq n \leq |\mathcal{D}| \right\} \tag{7}$$

Larger $\gamma_n$ values indicate samples with higher quality and significant contributions to reducing the downstream loss $J(\boldsymbol{\theta})$, especially when introduced during later training stages. In contrast, lower $\gamma_n$ values correspond to samples that are easy and less informative, better suited for early-stage training, or potentially noisy, which can be filtered out in data selection settings. Notably, our method calculates LQS on only a small number of data points, using a trained small model for scoring. Consequently, our method requires relatively low computational cost, as discussed in Appendix E.

### 3.3 LQS IMPLEMENTATION

Similar to (Gu et al., 2025), our data scoring method follows these steps: **1) Proxy data sampling**. A proxy dataset $\mathcal{D}^{\mathrm{prx}}$ is first uniformly sampled from the pre-training corpus $\mathcal{D}$, serving as a representative subset of the larger corpus. **2) Proxy data annotation.** We apply Eq. 7 to compute data scores for each instance in $\mathcal{D}^{\mathrm{prx}}$, obtaining a set of data samples with scores as ground truth (see Section 3.2). **3) Data scorer training.** The data scorer, typically a small LM, is fine-tuned on the automatically annotated data samples in $\mathcal{D}^{\mathrm{prx}}$ to predict data scores effectively. (see Section 3.2) **4) Full data scoring.** The trained data scorer is then applied to infer scores for the entire pre-training corpus $\mathcal{D}$.

**Proxy Data Annotation** To construct the ground truth scores $\boldsymbol{\gamma}^*$ for $\mathcal{D}^{\mathrm{prx}}$, we are inspired by (Gu et al., 2025) and adopt a bi-level optimization framework (see Algorithm 1) that quantifies the contribution of each data point in $\mathcal{D}^{\mathrm{prx}}$ to the downstream performance. The goal is to determine an optimal score vector $\boldsymbol{\gamma}^* = [\gamma_1^*, \gamma_2^*, \cdots, \gamma_{|\mathcal{D}|}^*]^\top$.

The bi-level optimization consists of two nested loops: 1) a forward loop that simulates the model training process, and 2) a reverse loop that adjusts the scores $\boldsymbol{\gamma}$ based on the model parameters at each training step. Specifically, in the **forward loop**, the model is trained for $T$ steps using gradient descent, where the training loss is weighted by the current scores $\boldsymbol{\gamma}^*$. This process updates the model parameters $\boldsymbol{\theta}_t$ iteratively, producing a trajectory of checkpoints $\boldsymbol{\theta}_t$ from $t = 0$ to $t = T - 1$. In the **reverse loop**, the target vector $\boldsymbol{\lambda}_t^*$ is computed through the training trajectory from $t = T-1$ to $t = 0$ according to Eq. 4, where $\boldsymbol{\lambda}_t^*$ represents the backward-propagated gradient at step $t$, $\nabla l(x_n, \boldsymbol{\theta}_t)$ is the gradient of the loss for the $n$-th data point. After each update, the scores are projected onto the probability simplex to ensure they remain valid probabilities $\mathrm{Proj}[\boldsymbol{\gamma}^*]$.

---

**Algorithm 1** Proxy Data Annotation

---

**Require:** LM learning rate $\eta$. Proxy data $\mathcal{D}^{\mathrm{prx}}$. Downstream loss $J(\boldsymbol{\theta})$. Training steps $T$. $\mathrm{Proj}[\cdot]$ that projects a point in $\mathbb{R}^{|\mathcal{D}|}$ to $U$. Model initialization $\boldsymbol{\theta}_0$.
**Ensure:** Data quality scores $\boldsymbol{\gamma}^*$.
$\quad \boldsymbol{\gamma}^* = [\gamma_1^*, \gamma_2^*, \cdots, \gamma_{|\mathcal{D}|}^*] \leftarrow \left[ \frac{1}{|\mathcal{D}^{\mathrm{prx}}|}, \frac{1}{|\mathcal{D}^{\mathrm{prx}}|}, \cdots, \frac{1}{|\mathcal{D}^{\mathrm{prx}}|} \right];$
$\quad$ **for** $t = 0, 1, \cdots, T - 1$ **do** $\hspace{4cm}$ ▷ Forward loop
$\quad\quad \boldsymbol{\theta}_{t+1} \leftarrow \boldsymbol{\theta}_t - \eta \nabla L(\boldsymbol{\theta}_t, \boldsymbol{\gamma})$
$\quad$ **end for**
$\quad \boldsymbol{\lambda}_T \leftarrow \nabla J(\boldsymbol{\theta}_T)$
$\quad$ **for** $t = T - 1, T - 2, \cdots, 1$ **do** $\hspace{3.3cm}$ ▷ Reverse loop
$\quad\quad \boldsymbol{\lambda}_t^* \leftarrow \boldsymbol{\lambda}_{t+1}^* + \nabla J(\boldsymbol{\theta}_t) - \eta \nabla^2 L(\boldsymbol{\theta}_t, \boldsymbol{\gamma}^*) \boldsymbol{\lambda}_{t+1}^* \hspace{1.5cm}$ ▷ Equation 4
$\quad$ **end for**
$\quad$ **for** $n = 1, 2, \cdots, |\mathcal{D}|$ **do**
$\quad\quad \gamma_n^* \leftarrow \gamma_n^* + \alpha \sum_{t=1}^{T-1} \frac{\boldsymbol{\lambda}_{t+1}^* {}^\top \nabla l(x_n, \boldsymbol{\theta}_t)}{\|\nabla l(x_n, \boldsymbol{\theta}_{t+1})\|} \hspace{3cm}$ ▷ Equation 7
$\quad$ **end for**
$\quad \boldsymbol{\gamma}^* \leftarrow \mathrm{Proj}[\boldsymbol{\gamma}^*]$
$\quad$ **return** $\boldsymbol{\gamma}^*$

---

**Data Scorer Training** After we obtain the scores $\boldsymbol{\gamma}^*$ for $\mathcal{D}^{\text{prx}}$, we then train a small LM, initialized from a pre-trained checkpoint, with a linear head to serve as the data scorer. The scorer is optimized on the proxy dataset $\mathcal{D}^{\text{prx}}$ to fit the ground-truth scores $\boldsymbol{\gamma}$ from Section 3.2. Specifically, each instance $x_n^{\text{prx}} \in \mathcal{D}^{\text{prx}}$ is encoded by averaging the LM's output hidden states along the sequence, producing a feature representation $\overline{\boldsymbol{h}}(x_n^{\text{prx}}, \boldsymbol{\phi}) \in \mathbb{R}^d$, where $\boldsymbol{\phi}$ are the LM parameters and $d$ is the hidden state size. This representation is passed through a linear head, with parameters $\boldsymbol{w} \in \mathbb{R}^d$ and $b \in \mathbb{R}$, to produce a predicted score. The parameters of the LM and linear head are optimized together using the Mean Squared Error (MSE) loss:

$$\mathcal{L}_{\text{MSE}} = \frac{1}{|\mathcal{D}^{\text{prx}}|} \sum_{n=1}^{|\mathcal{D}^{\text{prx}}|} \left( \boldsymbol{w}^\top \overline{\boldsymbol{h}}(x_n^{\text{prx}}, \boldsymbol{\phi}) + b - \gamma_n^* \right)^2, \tag{8}$$

The optimal parameters $\boldsymbol{\phi}^*$, $\boldsymbol{w}^*$, and $b^*$ are obtained by minimizing this loss. Once trained, the data scorer predicts scores for $x_n \in \mathcal{D}$ as $\gamma(x_n) = \boldsymbol{w}^{*\top} \overline{\boldsymbol{h}}(x_n, \boldsymbol{\phi}^*) + b^*$. This process enables the data scorer to generalize from $\mathcal{D}^{\text{prx}}$ to the larger pre-training corpus $\mathcal{D}$ effectively.

# 4 EXPERIMENT

## 4.1 EXPERIMENTAL SETUP

**Data.** (1) **General data.** We utilize the Redpajama (Weber et al., 2024) sourced from Common-Crawl as $\mathcal{D}$, which offers a relatively balanced knowledge distribution (Xie et al., 2024). The downstream loss $J(\theta)$ for the data scoring model is computed on the LIMA (Zhou et al., 2024), which is a high-quality dataset with 1,030 diverse instruction-response pairs spanning various downstream scenarios. (2) **Math data.** We use the OpenWebMath (Paster et al., 2023) as $\mathcal{D}$. The downstream loss $J(\theta)$ is computed on the MiniF2F (Zheng et al., 2021), which is a high-quality dataset consisting of 488 manually formalized mathematical problem statements, spanning multiple domains. (3) **Code data.** We employ The-Stack-v2 dataset (Lozhkov et al., 2024) as $\mathcal{D}$. The downstream loss $J(\theta)$ is computed on the Epicoder-380k Wang et al. (2025), which is a synthetic dataset with 380k diverse instruction-response pairs spanning multiple code generation scenarios.

**Model.** We apply the Mistral (Jiang et al., 2023) architecture for pre-training on general data, and the Qwen1.5 (Bai et al., 2023) for post-training in the math and code domain respectively using the official pre-trained weights.

**Baselines.** Based on the **DELT**, we compare our methods with baselines for data scoring and ordering, including: 1) Data scoring: Conventional (randomly shuffled data order without selection), KenLM (Heafield, 2011), DSIR (Xie et al., 2023), PDS (Gu et al., 2025); 2) Data ordering: Shuffling (Random), Sorting (Curriculum Learning) (Campos, 2021).

For additional details of the experimental setup, such as training and evaluation, see Appendix B.

## 4.2 MAIN RESULTS

**Data efficacy with different model sizes and data scales (Table 1).** We present the evaluation results of the pre-trained LMs on the OLMo (Groeneveld et al., 2024) evaluation benchmarks in Table 1. As shown, DELT consistently outperforms the baselines on most datasets, achieving the best overall performance across models with 160M, 470M, and 1B parameters (Table 1a), as well as across data sizes of 1B, 10B, and 50B tokens (Table 1b). We further validate the superior of our method on a larger model size (1.7B parameters) and a larger data scale (50B tokens). These results show that combining LQS and FO steadily enhances data efficacy in LM training across various model sizes, data scales, and downstream tasks. For more details, see Table 9 in Appendix.

**Data efficacy on different data ordering methods (Table 2).** Data ordering is a key component of the DELT and plays a significant role in improving data efficacy. To evaluate the performance of different ordering methods and demonstrate the superiority of our proposed FO, we conduct corresponding experiments. The results in Table 2 show that the proposed Folding method shows the most improvement among all the ordering methods. Besides, the ascending sorting improves the result while descending sorting leads to a decline.

**Data efficacy on different data scoring methods (Table 3).** Data scoring serves as the foundation for data ordering. To evaluate the performance of different data scoring methods on FO method, we

Table 1: Data efficacy results on different downstream benchmarks. The conventional method presents the average result over three random seeds in this and the following tables. DELT (Ours) means applying LQS for data scoring and FO for data ordering within the introduced paradigm.

(a) Results (%) for 1B-token data across model sizes (160M, 470M, 1B).

| | ARC-c | ARC-e | HS | LAMB | OBQA | PIQA | SciQ | Wino | Avg. |
|---|---|---|---|---|---|---|---|---|---|
| | | | | Model size = 160M | | | | | |
| Conventional | $21.27_{\pm 0.30}$ | $34.32_{\pm 0.83}$ | $27.85_{\pm 0.15}$ | $20.25_{\pm 1.70}$ | $24.40_{\pm 0.19}$ | $55.19_{\pm 0.16}$ | $56.93_{\pm 1.06}$ | $50.72_{\pm 0.84}$ | $36.37_{\pm 0.19}$ |
| DELT (Ours) | **21.59** | **36.07** | **28.41** | **23.79** | **25.60** | **56.37** | **59.80** | **53.04** | **38.08** |
| | | | | Model size = 470M | | | | | |
| Conventional | $21.16_{\pm 0.42}$ | $34.91_{\pm 0.86}$ | $28.11_{\pm 0.23}$ | $21.88_{\pm 0.10}$ | $23.90_{\pm 1.84}$ | $56.07_{\pm 0.58}$ | $58.75_{\pm 0.35}$ | $50.04_{\pm 1.40}$ | $36.85_{\pm 0.06}$ |
| DELT (Ours) | **22.33** | **35.88** | **28.45** | **23.26** | **26.60** | **57.20** | **60.10** | **52.81** | **38.33** |
| | | | | Model size = 1B | | | | | |
| Conventional | $20.58_{\pm 0.36}$ | $36.12_{\pm 0.03}$ | $28.32_{\pm 0.29}$ | $23.56_{\pm 0.77}$ | $25.00_{\pm 0.71}$ | $56.49_{\pm 1.19}$ | $60.05_{\pm 0.69}$ | **$52.07_{\pm 0.17}$** | $37.77_{\pm 0.07}$ |
| DELT (Ours) | **22.76** | **37.95** | **29.95** | **26.38** | **26.00** | **58.07** | **60.90** | 51.28 | **39.17** |

(b) Results (%) for 160M model across data sizes (10B, 50B).

| | ARC-c | ARC-e | HS | LAMB | OBQA | PIQA | SciQ | Wino | Avg. |
|---|---|---|---|---|---|---|---|---|---|
| | | | | Data size = 10B tokens | | | | | |
| Conventional | $22.82_{\pm 0.72}$ | $38.51_{\pm 0.57}$ | $30.72_{\pm 0.35}$ | $30.40_{\pm 0.77}$ | $25.70_{\pm 0.71}$ | $57.32_{\pm 0.27}$ | $64.90_{\pm 0.21}$ | $51.54_{\pm 0.95}$ | $40.24_{\pm 0.03}$ |
| DELT (Ours) | **24.38** | **39.80** | **31.64** | **32.98** | **27.21** | **58.56** | **66.70** | **51.67** | **41.62** |
| | | | | Data size = 50B tokens | | | | | |
| Conventional | $24.06_{\pm 0.06}$ | **$41.88_{\pm 0.06}$** | $32.05_{\pm 0.11}$ | $33.79_{\pm 2.18}$ | $26.80_{\pm 0.99}$ | $58.11_{\pm 0.58}$ | **$69.00_{\pm 1.06}$** | $51.93_{\pm 0.50}$ | $42.20_{\pm 0.13}$ |
| DELT (Ours) | **24.65** | 41.07 | **33.00** | **36.07** | **29.30** | **59.10** | 68.40 | **52.67** | **43.03** |

(c) Results (%) for 50B-token data and 1.7B model (large scale).

| | ARC-c | ARC-e | HS | LAMB | OBQA | PIQA | SciQ | Wino | Avg. |
|---|---|---|---|---|---|---|---|---|---|
| | | | | Model size = 1.7B & Data size = 50B tokens | | | | | |
| Conventional | $38.16_{\pm 1.09}$ | $48.02_{\pm 0.18}$ | $43.62_{\pm 0.47}$ | **$25.83_{\pm 0.38}$** | $41.00_{\pm 0.14}$ | $68.41_{\pm 0.38}$ | $65.70_{\pm 1.27}$ | **$68.93_{\pm 1.15}$** | $49.96_{\pm 0.21}$ |
| DELT (Ours) | **40.30** | **48.17** | **44.37** | 25.23 | **44.20** | **69.45** | **66.30** | 65.95 | **50.50** |

Table 2: Comparison of different data ordering methods on LQS-scored data. Sorting$_{asc}$ refers to **ascending** sorting by scores (low→high), while sorting$_{des}$ denotes **descending** sorting (high→low).

| Ordering | ARC-c | ARC-e | HS | LAMB | OBQA | PIQA | SciQ | Wino | Avg. |
|---|---|---|---|---|---|---|---|---|---|
| Conventional | 21.27 | 34.32 | 27.85 | 20.25 | 24.40 | 55.19 | 56.93 | 50.72 | 36.37 |
| Sorting$_{des}$ | 20.69 | 34.72 | 27.78 | 21.20 | 23.10 | 56.12 | 58.20 | 49.11 | 36.36↓ |
| Sorting$_{asc}$ | **22.18** | 35.40 | 28.01 | 23.48 | 23.80 | 55.60 | 56.80 | 51.07 | 37.04↑ |
| Folding | 21.59 | **36.07** | **28.41** | **23.79** | **25.60** | **56.37** | **59.80** | **53.04** | **38.08↑** |

Table 3: Comparison of different data scoring methods on FO-ordered data. Results are obtained on the full dataset without using any selection strategy.

| Scoring | ARC-c | ARC-e | HS | LAMB | OBQA | PIQA | SciQ | Wino | Avg. |
|---|---|---|---|---|---|---|---|---|---|
| Conventional | 21.27 | 34.32 | 27.85 | 20.25 | 24.40 | 55.19 | 56.93 | 50.72 | 36.37 |
| Kenlm | 20.98 | 35.00 | 28.02 | 22.55 | 23.90 | 56.54 | 58.30 | 51.36 | 37.08 |
| DSIR | **23.17** | 35.02 | **28.73** | 22.28 | 24.87 | **57.27** | 55.27 | 51.61 | 37.28 |
| PDS | 21.93 | 34.81 | 28.04 | 22.43 | **26.00** | 56.42 | 59.30 | 50.20 | 37.39 |
| LQS (Ours) | 21.59 | **36.07** | 28.41 | **23.79** | 25.60 | 56.37 | **59.80** | **53.04** | **38.08** |

conduct experiments to verify improvement in data efficacy. As shown in the table 3, applying FO led to consistent improvements in all scoring methods compared to the random baseline, with LQS achieving the greatest gains. For more related experiments, see Table 10 in Appendix.

Table 4: Comparison of various data scoring and ordering methods across different selection ratios.

| Scoring | Ordering | Selection Ratio | | | | | |
|---------|----------|------|------|------|------|------|------|
| | | 1.0 | 0.9 | 0.8 | 0.7 | 0.6 | 0.5 |
| Kenlm | Shuffle | 36.37 | 36.43 | 35.72 | 35.31 | 35.51 | 34.78 |
| | **Folding** | **37.08** | **37.22** | **36.55** | **36.38** | **36.16** | **35.44** |
| DSIR | Shuffle | 36.37 | 36.58 | 36.11 | 36.06 | 35.57 | 35.36 |
| | **Folding** | **37.13** | **37.33** | **36.90** | **36.70** | **36.20** | **36.08** |
| PDS | Shuffle | 36.37 | 36.19 | 36.24 | 37.01 | 36.14 | 36.03 |
| | **Folding** | **37.39** | **37.99** | **37.63** | **37.78** | **37.14** | **36.15** |
| **LQS** | Shuffle | 36.37 | 37.14 | 37.03 | 35.96 | 35.95 | 35.36 |
| | **Folding** | **38.08** | **37.20** | **37.54** | **37.58** | **36.60** | **36.47** |

**Data efficacy on different data selection ratios (Table 4).** To evaluate the performance of our methods across different data selection ratios, we applied FO and LQS under varying selection ratios and compared them with scoring methods from advanced data efficiency baselines. As shown in Table 4, the FO method consistently demonstrates stable improvements across different scoring methods and selection ratios. For the LQS method, it outperforms other baselines under the random shuffle setting across most selection ratios. The combination of LQS and FO achieves the highest average accuracy of 38.08% when the selection ratio is 1.0 and shows superior performance across most selection ratios. For more details on the performance of various combinations under different selection ratios, please refer to Table 7, Table 12, and Figure 11 in Appendix.

**Domain Robustness in Post-training (Table 5).** To validate the robustness of the DELT paradigm, we conduct post-training experiments on datasets of OpenWebMath (Paster et al., 2023) and The-Stack-v2 (Lozhkov et al., 2024) respectively across math and code domains, both of which are sourced from the web data. As shown in Table 5, our method consistently outperforms the baselines across benchmarks in different domains, demonstrating the strong versatility of the DELT under our proposed LQS and FO methods.

Table 5: Data efficacy of models trained on different domain-specific datasets.

(a) Results on code domain.

| | Method | HumanEval | MBPP |
|---|--------|-----------|------|
| Qwen1.5-0.5B | Conventional | 7.00 | 7.93 |
| | Ours | **9.76** | **9.40** |
| Qwen1.5-1.8B | Conventional | 9.15 | 12.00 |
| | Ours | **16.46** | **13.20** |

(b) Results on math domain.

| | Method | MathQA | GPQA Diamond |
|---|--------|--------|--------------|
| Qwen1.5-0.5B | Conventional | 21.23 | 24.92 |
| | Ours | **22.73** | **26.83** |
| Qwen1.5-1.8B | Conventional | 22.72 | 27.17 |
| | Ours | **24.75** | **28.94** |

**Stability on Different Epochs (Figure 4).** In addition to one-epoch training, we also evaluate the impact of DELT under a multi-epoch setting. As shown in Figure 4, our method consistently boosts the results of conventional random ordering as the number of epochs increases. While conventional random ordering exhibits fluctuations and slow improvements after the second epoch, our approach demonstrates steady progress, showcasing the effectiveness and stability of our method in maintaining superior performance over multiple epochs. For more details, see Table 13 in Appendix.

## 4.3 ABLATION STUDY

$L$ **in Folding Learning (Figure 5).** We explore the influence of different folding layers $L$ on model performance in Figure 5. As $L > 1$, the performance consistently surpasses that of $L = 1$ (curriculum learning), verifying the benefits brought by folding learning. The average performance initially increases and then gradually declines, peaking at $L = 3$. In the experiments conducted in this paper, $L$ is set to a default value of 3. For more results and analyses on FO, please refer to Table 14 and Table 15 in the Appendix, respectively.

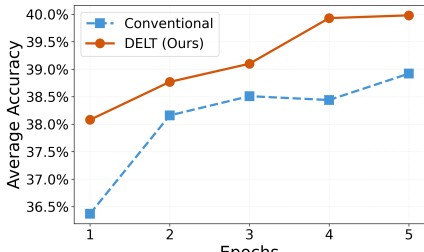 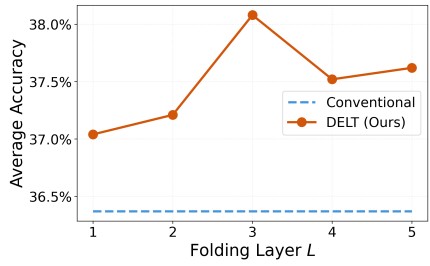

Figure 4: Performance on different epochs.     Figure 5: Influence of the folding layers $L$.

## 5 CONCLUSION

To address the underexplored research on data efficacy, we propose a novel data ordering method, Folding Ordering (FO), and a new data scoring method, Learnability-Quality Scoring (LQS). To further advance data efficacy, we develop the DELT paradigm, which integrates the proposed data ordering and data scoring modules to enhance data efficacy. Additionally, DELT optionally incorporates data selection module to improve data efficiency as well. Our comprehensive experiments with various DELT implementations confirm its effectiveness, demonstrating that our newly designed methods for data scoring and ordering outperform existing approaches. We believe that the proposed methods and paradigm highlight the significant potential of data efficacy, which focuses on improving model performance without increasing data scale or model scale.

ETHICS STATEMENT

This work focuses on improving data efficacy for LM training. While our model is trained on standard open-source datasets and tested in controlled settings, we acknowledge that any AI system may potentially exhibit biases or produce unexpected behaviors. Our research is intended for academic exploration only, and we emphasize that any such outcomes do not reflect the views of the authors. We support the development of AI technologies that are ethical, safe, and aligned with societal values.

REPRODUCIBILITY STATEMENT

This section discusses the authors' efforts to maximize the reproducibility of the methods and implementations presented, from the following two perspectives. First, we provide complete, runnable code in the supplementary material with detailed execution instructions. The provided code covers all experimental pipelines and settings in our paper, including data collection, scoring, ordering, and selection, as well as model training and evaluation.

Secondly, we have provided detailed explanations of the implementation to enhance the reproducibility of our method: 1) Algorithm Implementation. In Appendix 3.2 and E, we present a comprehensive discussion of the implementation details of our method, along with pseudocode. 2) Model Training and Testing. In Section 4.1, we provide detailed information about the data, models, and baseline methods. Additionally, in Appendix B, we elaborate on all experimental setups, training details, computational resources, and other relevant information.

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

## APPENDIX

## A  DECLARATION OF LLM USAGE

In the preparation of this work, the authors used LLM (e.g., GPT-4) in order to improve the readability and language of the manuscript. After using this tool, the authors reviewed and edited the content as needed and take full responsibility for the content of the published article.

## B  ADDITIONAL EXPERIMENTAL SETUP

**Training.** Unless otherwise specified, we pre-train all LMs for one epoch, using a batch size of 512 and a maximum input length of 1,024. For the model, we utilize 160M parameters by default. For pre-training, we randomly select a 1B-token subset from Redpajama by default, while for post-training, we sample a 1B-token subset each from OpenWebMath and The-Stack-v2.

**Evaluation.** For general evaluation, we assess the trained models on a range of standard natural language understanding and reasoning benchmarks, including Hellaswag (HS; Zellers et al., 2019), Winogrande (Wino; Levesque et al., 2012), LAMBADA (LAMB; Paperno et al., 2016), OpenbookQA (OBQA; Mihaylov et al., 2018), ARC-easy/challenge (ARC-e/c; Clark et al., 2018), PIQA (Bisk et al., 2020), SciQ (Welbl et al., 2017), and BoolQ (Clark et al., 2019). For domain-specific tasks, we assess the models on mathematical reasoning and code generation benchmarks. Specifically, for math, we use GPQA Diamond (Rein et al., 2024) and MathQA (Amini et al., 2019), while for code, we use HumanEval (Chen et al., 2021) and MBPP (Austin et al., 2021).

For the code generation benchmarks, we use a 0-shot setting for HumanEval and a 3-shot setting for MBPP. Code generation performance is reported using pass rate@1, which indicates the percentage of first-attempt solutions that pass all associated unit tests (pass@1). For the other benchmarks,

the tasks are framed as multiple-choice questions, where the model selects the correct answer by minimizing the normalized loss across all candidate options (acc_norm).

**Compute Resources.** We train the 160M model on 1B-token datasets using a single NVIDIA A100 40GB GPU. For experiments with the 160M, 470M, and 1B models on 10B and 50B-token datasets, we utilize 8 NVIDIA A100 40GB GPUs. All data scoring steps, including proxy data annotation, scorer training, and data scoring, are performed on a single NVIDIA A100 80GB GPU.

**Training Configuration.** To create the data scorer, we fine-tune the Fairseq-Dense-125M model (Artetxe et al., 2022) on the solved data weights $\gamma$. As described in algorithm 1, we apply a linear transformation to the mean-pooled representations of instances along the sequence length. The hidden state size is set to 768. The optimization of algorithm 1 is performed using the AdamW optimizer (Loshchilov & Hutter, 2019) for 5 epochs, with a learning rate of $1 \times 10^{-4}$ and a batch size of 512. We train on 90% of the samples and reserve the remaining 10% from $\mathcal{D}^{\mathrm{prx}}$ as a validation set. The checkpoint with the highest Spearman correlation score (De Winter et al., 2016; Gu et al., 2025) on the validation set is selected to infer data quality scores in $\mathcal{D}$.

For LMs training, all models are trained with a batch size of 256 and a maximum input sequence length of 1,024 for one epoch. The AdamW optimizer (Loshchilov & Hutter, 2019) is paired with a cosine learning rate scheduler. The scheduler includes a warm-up phase for the first 2,000 steps, after which the learning rate decays to 10% of its peak value. The model architecture and corresponding learning rates are summarized in Table 6, following the configurations in (Gu et al., 2025).

| Model Size | $d_{\mathrm{model}}$ | $d_{\mathrm{FFN}}$ | $n_{\mathrm{layers}}$ | $n_{\mathrm{head}}$ | $d_{\mathrm{head}}$ | learning rate |
|---|---|---|---|---|---|---|
| 160M | 768 | 3,072 | 12 | 12 | 64 | $6 \times 10^{-4}$ |
| 470M | 1,024 | 4,096 | 24 | 16 | 64 | $3 \times 10^{-4}$ |
| 1B | 1,536 | 6,144 | 24 | 16 | 96 | $2.5 \times 10^{-4}$ |
| 1.7B | 2,048 | 8,192 | 24 | 16 | 128 | $2 \times 10^{-4}$ |

Table 6: Model configurations and corresponding learning rates.

To provide a clearer overview of our experimental setup, we present the following table for reference. Table 7 details the goal, data, and model settings for all experiments conducted in the main text.

## C  LIMITATIONS AND FUTURE WORK

The proposed paradigm has two primary limitations. 1) The current verification is specifically focused on language models, with no evaluation in other modalities like image and audio, which depend on different scoring implementations. 2) Similar to PDS data scoring, the implementation of our designed LQS method requires calculating the downstream loss $J(\theta)$ on a high quality and small scale dataset. In the future, we plan to scale up our method on larger models (e.g., tens or hundreds of billions of parameters) and larger datasets (terabyte level). Additionally, we aim to explore simpler and more effective data scoring methods and extend this paradigm to multimodal models.

## D  DELT PARADIGM

To lay the foundation for the further development of data efficacy, we propose a paradigm (Figure 6) to enhance **data efficacy** in LM training without modifying the data content $\mathcal{D}$ or the model parameters $\theta$. It currently comprises two components:

- **Data Scoring**: it aims to assign a score for each training sample based on specific criteria, such as quality, difficulty, diversity, and learnability. These scores are then applied to guide data selection and data ordering in subsequent stages.
- **Data Ordering**: it targets at reorganizing the order of training samples in $\mathcal{D}$ (or $\mathcal{D}^{\mathrm{sub}}$) to create $\mathcal{D}'$, such that LMs trained on $\mathcal{D}'$ achieve superior performance. This process focuses on the organization of dataset $\mathcal{D}$ (or $\mathcal{D}^{\mathrm{sub}}$), while it does not change the dataset scale.

Table 7: Detailed experimental setting.

| Category | Table 1 (a) | Table 1 (b) | Table 1 (c) | Table 2 | Table 3 | Table 4 | Table 5 (a) | Table 5 (b) |
|---|---|---|---|---|---|---|---|---|
| **Goal** | Robustness across different model sizes (Pre-train) | Robustness across different data scales (Pre-train) | Scalability to relatively large model size and data scale (Pre-train) | Comparison of different ordering methods using the same scoring method (LQS) (Pre-train) | Comparison of different scoring methods using the same ordering method (FO) (Pre-train) | Robustness across different data selection ratios (Pre-train) | Robustness on the Math domain (Post-train) | Robustness on the Code domain (Post-train) |
| **Data** | 1B-token data from RedPajama | 10B-token, 50B-token data from RedPajama | 50B-token data from RedPajama | 1B-token data from RedPajama | 1B-token data from RedPajama | Subsets of the 1B-token RedPajama data based on selection ratios from 0.5 to 1.0. | 1B-token data from Open-Web-Math | 1B-token data from The Stack v2 |
| **Model** | 160M, 470M, 1B Mistral | 160M Mistral | 1.7B Mistral | 160M Mistral | 160M Mistral | 160M Mistral | Qwen1.5-0.5B, 1.8B | Qwen1.5-0.5B, 1.8B |

In addition, to make the paradigm more general, we also integrate an optional **data selection** module, enabling DELT to seamlessly enhance **data efficiency**. This module selects an optimal subset $\mathcal{D}^{\text{sub}}$ from the dataset $\mathcal{D}$, ensuring that LMs trained on $\mathcal{D}^{\text{sub}}$ achieve the best possible performance. While this process adjusts the size of the dataset $\mathcal{D}$, it does not affect the structure of $\mathcal{D}^{\text{sub}}$.

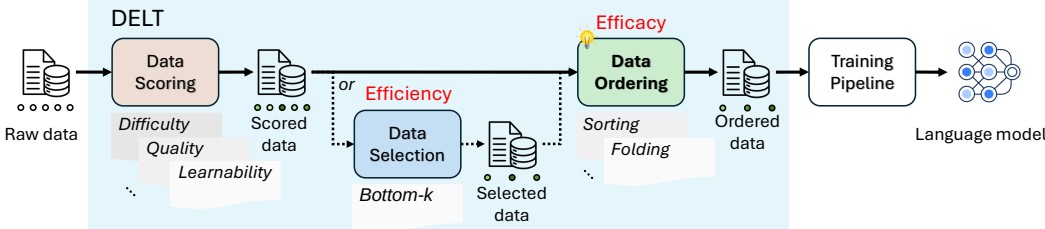

Figure 6: Paradigm of Data Efficacy for LM training. The blue box represents the paradigm DELT. Both the methods for data scoring and data ordering components in DELT are flexible and can be adjusted, including those outlined in Section 3. Meanwhile, the data selection is an optional component that can further improve data efficiency. Within the proposed DELT, data efficacy and efficiency are seamlessly compatible, working together to optimize language model performance.

Unlike the baseline method, where the language model is trained directly on the raw data $\mathcal{D}$, and the data efficiency methods that use a selected subset $\mathcal{D}^{\text{sub}}$, DELT processes the raw data $\mathcal{D}$ as follows:

Firstly, data scoring, defined as $f$, assigns a score vector $\boldsymbol{\gamma}$ to the raw data $\mathcal{D}$, where $\boldsymbol{\gamma}$ lies in a $|\mathcal{D}|$-dimensional simplex. Samples with large $\boldsymbol{\gamma}$ are considered good according to their criteria.

$$\boldsymbol{\gamma} = f(\mathcal{D}) = \left[\gamma_1, \gamma_2, \cdots, \gamma_{|\mathcal{D}|}\right]^{\top} \tag{9}$$

Secondly, the optioned data selection, denoted as $f_s$, identifies a subset $\mathcal{D}^{\text{sub}}$ from $\mathcal{D}$ based on the scores $\boldsymbol{\gamma}$ by the selection ratio $r$. The number of samples $K$ to be selected is determined by $r$. The function $rank$ provides the ranking index of each element in the set $\boldsymbol{\gamma}$ in ascending order.

$$\mathcal{D}^{\text{sub}} = f_s(\mathcal{D}; \boldsymbol{\gamma}, K) = \{x_k \mid rank(\boldsymbol{\gamma}_k) > |\mathcal{D}| - K \text{ and } 1 \le k \le |\mathcal{D}|\} \tag{10}$$

$$K = \lfloor r \cdot |\mathcal{D}| \rfloor \tag{11}$$

Finally, data ordering, represented by $f_o$, reorganizes the $\mathcal{D}$ or $\mathcal{D}^{\text{sub}}$ into a new dataset $\mathcal{D}'$ with unchanged size, based on a permutation $\pi$ determined by $\boldsymbol{\gamma}$. It could be $\pi_{sort}$ that returns the indices of each element in $\boldsymbol{\gamma}$ after sorting or other functions.

$$\mathcal{D}' = f_o(\mathcal{D}; \boldsymbol{\gamma}) = \left[x_{\pi(\boldsymbol{\gamma})_1}, x_{\pi(\boldsymbol{\gamma})_2}, \cdots, x_{\pi(\boldsymbol{\gamma})_{|\mathcal{D}|}}\right] \tag{12}$$

**Compatibility of data efficacy and data efficiency in DELT.** As shown in Figure 6, the DELT paradigm can build upon data scoring and data ordering by incorporating data selection to further enhance **data efficiency**. The entire DELT process can be defined as a transformation of the original dataset $\mathcal{D}$ into a reordered dataset $\mathcal{D}'$:

$$\mathcal{D}' = f_o(\boldsymbol{\gamma}_o) \circ f_s(\mathcal{D}; \boldsymbol{\gamma}_s, K), \tag{13}$$

where the symbol $\circ$ denotes functional composition. $\boldsymbol{\gamma}_o$ and $\boldsymbol{\gamma}_s$ are the score vectors for data ordering and data selection, respectively. Since data scoring often requires substantial computation time, both data selection and data ordering in DELT apply a shared score vector for practicality, i.e., $\boldsymbol{\gamma}_o = \boldsymbol{\gamma}_s = \boldsymbol{\gamma}$. This process ensures that the most qualified samples are selected and then optimally ordered, thereby significantly improving model performance in both data efficacy and efficiency.

## E  LQS COMPUTATIONAL ANALYSIS.

We compare the computational complexity of LQS with pre-training in Table 8. Specifically, following Hoffmann et al. (2022), for an LM with $N$ parameters to be trained on $D$ tokens, we assume

the computational FLOPs of a forward and a backward pass are $2ND$ and $4ND$, respectively. We compute the FLOPs and asymptotic complexities of different stages in LQS as follows.

**Proxy Data Annotation.** According to Section Appendix E, we first pre-train a proxy LM on $D$ which consumes $6N^{prx}D$ FLOPs. Then, we perform Algorithm 1 in our paper $M$ times on $D^{prx}$ based on the proxy LM. The forward inner loop in Algorithm 1 consumes $6N^{prx}D^{prx}$ FLOPs. The reverse loop can be treated as the "backward" propagation of the forward inner loop as discussed in Appendix E.1, which consumes $2 \times 6N^{prx}D^{prx}$ FLOPs. The update of $\gamma$ results in one forward and backward pass of the proxy LM on $D^{prx}$, which consumes $6N^{prx}D^{prx}$ FLOPs. In summary, the asymptotic complexity of proxy data annotation (LQS calculation) is $O(N^{prx}D + 4MN^{prx}D^{prx})$.

**Data Scorer Training.** The data scorer with $N^{score}$ is trained on $D^{prx}$ and used to infer data quality scores on $D$. Therefore, the computation overhead is $6N^{score}D^{prx} + 2N^{score}D$ and the asymptotic complexity is $O(3N^{score}D^{prx} + N^{score}D)$.

**Data Selection & Ordering.** Selecting and ordering pre-training corpus requires iterating over $D$, whose asymptotic complexity is $O(D)$. This process can be done on CPUs and does not require GPU FLOPs.

**Language Model Training.** Pre-training an LM with $N$ parameters requires $6ND$ FLOPs, whose asymptotic complexity is $O(ND)$.

Overall, since the parameter size of the LM used for training $N$, and the data size $D$, are typically much larger than the parameter size of the data-scoring model $N^{score}$, and the proxy data size $D^{prx}$, the ***computational cost of model training is significantly higher than that of sample scoring***.

Table 8: Asymptotic complexity, GPU FLOPs, and actually spent time of different DELT steps and 1B model pre-training.

| Step | Complexity | FLOPs ($\times 10^{20}$) | Actual Time |
|------|-----------|--------------------------|-------------|
| Proxy Data Annotation (LQS) | $O(N_{\text{prx}}D + 4MN_{\text{prx}}D_{\text{prx}})$ | 0.49 | 10.2 Hours |
| Data Scorer Training & Infering (LQS) | $O(3N_{\text{score}}D_{\text{prx}} + N_{\text{score}}D)$ | 0.063 | 1.50 Hours |
| Data Selection & Ordering | $O(D)$ | 0 | 1.3 Minutes |
| LM Training | $O(ND)$ | 2.8 | 96 Hours |

# F MORE EXPERIMENTS RESULTS

## F.1 ROBUSTNESS EXPERIMENTS

**Data Efficacy across Different Model Sizes and Data Scales on More Methods. (Table 9).** To supplement Table 1, we provide additional experimental results comparing DELT with data selection methods across different model sizes and data scales. As shown in Table 9, with increasing model parameters and data scales, our proposed DELT demonstrates consistent improvements and outperforms various methods across most benchmarks.

**Robustness of folding ordering method.** To complement Table 2, we rigorously validate the effectiveness and generality of our folding ordering (FO) strategy by applying it to two strong data scoring baselines: Fineweb-edu (Penedo et al., 2024) and QuRating (Wettig et al., 2024).

- **Fineweb-edu** (Penedo et al., 2024) targets the educational value and learnability of web text and provides publicly available scores at web scale.
- **QuRating** (Wettig et al., 2024) elicits pairwise comparisons from GPT-3.5-turbo on four criteria (writing style, facts and trivia, educational value, and required expertise) and then aggregates these signals with a Bradley–Terry model to produce a single scalar score for each document.

In our experiments, we directly reuse their open scores on the CommonCrawl subset and randomly sample 1B-tokens for training: we take the `score` field from Fineweb-edu [1] and the

---
[1] https://huggingface.co/datasets/HuggingFaceFW/fineweb-edu

Table 9: Efficacy results on different downstream benchmarks. Ours means applying LQS for data scoring and Folding for data ordering within the DELT paradigm.

(a) Results (%) for 1B-token data across model sizes (160M, 470M, 1B).

| | ARC-c | ARC-e | HS | LAMB | OBQA | PIQA | SciQ | Wino | Avg. |
|---|---|---|---|---|---|---|---|---|---|
| | | | | Model size = 160M | | | | | |
| Conventional | 21.27 | 34.32 | 27.85 | 20.25 | 24.40 | 55.19 | 56.93 | 50.72 | 36.37 |
| KenLM | **21.93** | 33.96 | 28.09 | 20.69 | 25.20 | 54.79 | 56.20 | 50.59 | 36.43 |
| PDS | 21.84 | 35.02 | 27.61 | 19.93 | 24.80 | **56.23** | 59.00 | 51.38 | 37.01 |
| Ours | 21.59 | **36.07** | **28.41** | **23.79** | **25.60** | 56.37 | **59.80** | **53.04** | **38.08** |
| | | | | Model size = 470M | | | | | |
| Conventional | 21.16 | 34.91 | 28.11 | 21.88 | 23.90 | 56.07 | 58.75 | 50.04 | 36.85 |
| KenLM | **22.35** | 34.85 | 28.05 | 20.51 | 25.00 | 55.17 | 56.60 | 50.04 | 36.57 |
| PDS | 22.10 | 33.04 | 27.84 | 21.25 | 24.80 | 56.96 | 59.80 | 51.85 | 37.23 |
| Ours | 22.33 | **35.88** | **28.45** | **23.26** | **26.60** | **57.20** | **60.10** | **52.81** | **38.33** |
| | | | | Model size = 1B | | | | | |
| Conventional | 20.58 | 36.12 | 28.32 | 23.56 | 25.00 | 56.49 | 60.05 | 52.07 | 37.77 |
| KenLM | 21.67 | 35.86 | 28.76 | 23.46 | **26.80** | 56.58 | 59.00 | 49.88 | 37.75 |
| PDS | 22.10 | 35.56 | 28.20 | 23.56 | 26.40 | 56.37 | 60.50 | 50.67 | 37.92 |
| Ours | **22.76** | **37.95** | **29.95** | **26.38** | 26.00 | **58.07** | **60.90** | 51.28 | **39.17** |

(b) Results (%) for 160M model across data sizes (10B, 50B).

| | ARC-c | ARC-e | HS | LAMB | OBQA | PIQA | SciQ | Wino | Avg. |
|---|---|---|---|---|---|---|---|---|---|
| | | | | Data size = 10B tokens | | | | | |
| Conventional | 22.82 | 38.51 | 30.72 | 30.40 | 25.70 | 57.32 | 64.90 | 51.54 | 40.24 |
| KenLM | 22.78 | 37.92 | 30.54 | 29.98 | 25.60 | 57.29 | 66.00 | 52.80 | 40.36 |
| PDS | 22.70 | 39.35 | 30.73 | 31.85 | 27.20 | 56.04 | 64.90 | **52.88** | 40.71 |
| Ours | **24.38** | **39.80** | **31.64** | **32.98** | **27.21** | **58.56** | **66.70** | 51.67 | **41.62** |
| | | | | Data size = 50B tokens | | | | | |
| Conventional | 24.06 | **41.88** | 32.05 | 33.79 | 26.80 | 58.11 | **69.00** | 51.93 | 42.20 |
| KenLM | 23.74 | 40.14 | 32.10 | 35.13 | 28.41 | 58.15 | 67.52 | 51.71 | 42.11 |
| PDS | 24.57 | 41.37 | 32.44 | 35.36 | 29.20 | **59.25** | 68.10 | 50.83 | 42.64 |
| Ours | **24.65** | 41.07 | **33.00** | **36.07** | **29.30** | 59.10 | 68.40 | **52.67** | **43.03** |

educational_value_average field from QuRating [2]. As shown in Table 10, Applying FO on these scores yields consistent and substantial improvements across evaluations, demonstrating that FO is robust and broadly applicable to diverse data scoring signals.

Table 10: Performance of FO on different data scoring methods with different data source.

|  | Ordering | ARC-c | ARC-e | HS | LAMB | OBQA | PIQA | SciQ | Wino | Avg. |
|---|---|---|---|---|---|---|---|---|---|---|
| Fineweb-edu | Shuffle | 22.53 | 35.44 | **27.87** | **14.63** | 24.40 | 56.37 | 53.50 | **51.54** | 35.78 |
|  | **Folding** | **23.46** | **38.64** | 27.71 | 10.58 | **27.60** | **56.47** | **61.70** | 50.36 | **37.07** |
| QuRating | Shuffle | 22.01 | 33.42 | 27.60 | **18.20** | **24.40** | 56.15 | **57.00** | 51.22 | 36.25 |
|  | **Folding** | **23.55** | **34.76** | **28.16** | 17.52 | 24.00 | **57.73** | 55.80 | **53.20** | **36.84** |

**Comparison of Data Efficiency (Table 11).** To supplement Table 4, we further highlight the superiority of the DELT framework by comparing it with the data efficiency setting. Table 11 presents the results of different methods applied within the DELT paradigm, all of which significantly outperform the conventional baseline. Notably, regardless of whether data selection is applied, our proposed LQS scoring method achieves the best results. Furthermore, our proposed Folding ordering method consistently provides noticeable improvements across all baseline methods.

Table 11: Efficacy results of different DELT implementations. The best scores for each model size are highlighted in **bold**, while the second-best scores are shown in ***italic bold***. The selection methods report the highest scores across all selection ratios.

| Pipeline | Scoring | Selection | Ordering | ARC-c | ARC-e | HS | LAMB | OBQA | PIQA | SciQ | Wino | Avg. |
|---|---|---|---|---|---|---|---|---|---|---|---|---|
| Conventional | - | - | - | 21.27 | 34.32 | 27.85 | 20.25 | 24.40 | 55.19 | 56.93 | 50.72 | 36.37 |
| DELT | KenLM | - | Sorting | 21.93 | 33.96 | ***28.09*** | 20.69 | 25.20 | 54.79 | 56.20 | 50.59 | 36.43 |
|  | KenLM | - | Folding | 20.98 | 35.00 | 28.02 | 22.55 | 23.90 | ***56.54*** | 58.30 | ***51.36*** | 37.08 |
|  | PDS | - | Sorting | ***22.44*** | 34.18 | 27.98 | 21.35 | 25.40 | 55.28 | 55.80 | 49.17 | 36.45 |
|  | PDS | - | Folding | 21.93 | 34.81 | 28.04 | 22.43 | **26.00** | 56.42 | ***59.30*** | 50.20 | 37.40 |
|  | LQS | - | Sorting | **23.22** | ***35.24*** | 28.03 | ***22.79*** | 24.70 | ***56.85*** | 57.90 | 51.17 | ***37.49*** |
|  | LQS | - | Folding | 21.59 | **36.07** | **28.41** | 23.79 | ***25.60*** | 56.37 | **59.80** | 53.04 | **38.08** |
|  | KenLM | ✓ | Sorting | 21.93 | 34.68 | 27.78 | 19.37 | **26.40** | 54.95 | 56.30 | ***52.96*** | 36.80 |
|  | KenLM | ✓ | Folding | ***22.10*** | 34.30 | 27.62 | 21.56 | 25.00 | 56.26 | 58.10 | 52.80 | 37.22 |
|  | PDS | ✓ | Sorting | 22.61 | 35.27 | ***28.08*** | 19.68 | ***25.80*** | 56.53 | 59.60 | 51.54 | 37.38 |
|  | PDS | ✓ | Folding | 21.66 | ***36.01*** | 28.05 | 24.33 | 24.10 | 55.61 | **61.70** | 52.47 | ***37.99*** |
|  | LQS | ✓ | Sorting | ***22.10*** | 35.61 | 28.05 | 22.53 | 23.60 | 55.93 | 59.60 | 51.38 | 37.35 |
|  | LQS | ✓ | Folding | 21.59 | **36.07** | **28.41** | ***23.79*** | 25.60 | ***56.37*** | **59.80** | 53.04 | **38.08** |

**Comparison with advanced data selection methods.** To supplement Table 4, we presents the results comparing different data scoring methods without data ordering in Table 12. Compared to the results from the conventional pipeline and state-of-the-art data efficiency baselines, the DELT framework, which incorporates both data selection and ordering, consistently demonstrates significant improvements across various method combinations. Notably, our LQS also achieves the best results even in the setting where only data selection is applied.

Table 12: Comparison among different data scoring methods. The selection methods report the highest scores across all selection ratios.

| Pipeline | Scoring | Selection | Ordering | ARC-c | ARC-e | HS | LAMB | OBQA | PIQA | SciQ | Wino | Avg. |
|---|---|---|---|---|---|---|---|---|---|---|---|---|
| Conventional | - | - | - | 21.27 | 34.32 | 27.85 | 20.25 | 24.40 | 55.19 | 56.93 | 50.72 | 36.37 |
| Efficiency | KenLM | ✓ | - | 21.42 | 34.34 | 27.76 | 20.84 | 25.00 | 56.31 | 54.30 | 51.07 | 36.38 |
|  | DSIR | ✓ | - | **22.69** | 34.55 | 28.26 | 21.81 | 24.40 | **56.80** | 54.80 | 51.14 | 36.81 |
|  | PDS | ✓ | - | 21.84 | 35.02 | 27.61 | 19.93 | 24.80 | 56.23 | 59.00 | 51.38 | 37.01 |
|  | LQS (Ours) | ✓ | - | 22.18 | 34.09 | 27.80 | 21.02 | 25.20 | 55.98 | 59.00 | 51.85 | 37.14 |
| DELT | LQS (Ours) | ✓ | Folding | 21.59 | **36.07** | **28.41** | **23.79** | **25.60** | 56.37 | **59.80** | **53.04** | **38.08** |

**Data efficiency promotion on existing methods (Figure 7).** To supplement Table 4, we further evaluates the data efficiency of the DELT framework by involving the data selection setting in Figure 7. Compared to the random selection baseline (Shuffling), the DELT framework (Sorting and Folding) achieves superior performance across the majority of selection ratios. The results show

---

[2]https://huggingface.co/datasets/princeton-nlp/QuRatedPajama-260B

that the DELT framework is compatible with the data selection method, and the combination further improves their data efficiency.

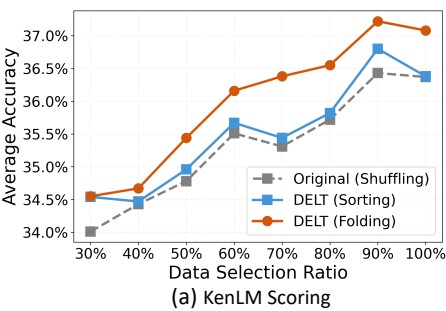 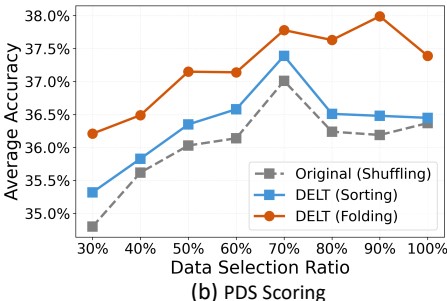

(a) KenLM Scoring      (b) PDS Scoring

Figure 7: The performance of KenLM (Heafield, 2011) and PDS (Gu et al., 2025) under different data selection ratios, both with and without the DELT paradigm. Data efficiency is enhanced when integrated into DELT.

**Stability on Different Epochs. (Table 13)** To supplement Figure 4, we further report the detailed results of the proposed DELT across different epochs on various benchmarks. As shown in Table 13, with an increasing number of epochs, our method demonstrates stable improvements across most benchmarks, further highlighting its robustness and generalizability.

Table 13: Results on OLMo for the different epochs.

| Epoch | | ARC-c | ARC-e | HS | LAMB | OBQA | PIQA | SciQ | Wino | Avg. |
|---|---|---|---|---|---|---|---|---|---|---|
| 1 | Conventional | 21.27 | 34.32 | 27.85 | 20.25 | 24.40 | 55.19 | 56.93 | 50.72 | 36.37 |
| | DELT (Ours) | **21.59** | **36.07** | **28.41** | **23.79** | **25.60** | **56.37** | **59.80** | **53.04** | **38.08** |
| 2 | Conventional | 21.93 | 36.20 | **29.18** | 25.93 | 23.00 | **56.86** | **61.20** | 50.99 | 38.16 |
| | DELT (Ours) | **22.35** | **36.41** | 28.28 | **27.63** | **26.80** | 56.47 | 61.00 | **51.22** | **38.77** |
| 3 | Conventional | 21.35 | 35.78 | 28.76 | 27.14 | **26.20** | **56.51** | **62.80** | 49.51 | 38.51 |
| | DELT (Ours) | **22.44** | **36.95** | **29.41** | **29.09** | 24.80 | 56.20 | 62.30 | **51.62** | **39.10** |
| 4 | Conventional | 21.10 | 35.99 | 28.97 | 27.51 | 27.20 | 55.69 | 61.80 | 49.28 | 38.44 |
| | DELT (Ours) | **22.53** | **38.05** | **29.78** | **29.58** | 26.40 | **57.34** | **63.90** | **51.85** | **39.93** |
| 5 | Conventional | 20.59 | 37.55 | 29.31 | 28.05 | **27.00** | 57.11 | 61.20 | **50.54** | 38.92 |
| | DELT (Ours) | **22.87** | **38.05** | **30.01** | **30.08** | 26.80 | **58.16** | **64.10** | 49.80 | **39.98** |

**Details for $L$ in Folding Learning (Table 14).** To supplement Figure 5, we provide detailed results for the proposed Folding Learning method with varying values of the parameter $L$. As shown in Table 14, model performance reaches its peak at $L = 3$ and demonstrates significant advantages across most benchmarks. Notably, compared to traditional Curriculum Learning ($L = 1$), our proposed method ($L > 1$) achieves substantially better performance on all benchmarks.

Table 14: Effect of the fold layer $L$. $L = -$ represents the conventional method, which is three times the random average results. When $L = 1$, the ordering method reduces to curriculum learning.

| $L$ | ARC-c | ARC-e | HS | LAMB | OBQA | PIQA | SciQ | Wino | Avg. |
|---|---|---|---|---|---|---|---|---|---|
| - | 21.27 | 34.32 | 27.85 | 20.25 | 24.40 | 55.19 | 56.93 | 50.72 | 36.37 |
| 1 | 22.18 | 35.40 | 28.01 | 23.48 | 23.80 | 55.60 | 56.80 | 51.07 | 37.04 |
| 2 | 21.57 | 34.26 | 28.34 | 23.29 | 25.80 | 55.88 | 58.70 | 49.80 | 37.21 |
| 3 | 21.59 | **36.07** | **28.41** | **23.79** | 25.60 | 56.37 | **59.80** | **53.04** | **38.08** |
| 4 | 22.83 | 34.98 | 28.50 | 22.35 | 24.90 | **56.67** | 59.80 | 50.10 | 37.52 |
| 5 | **22.91** | 35.57 | 28.16 | 22.85 | **26.70** | 55.41 | 57.30 | 52.08 | 37.62 |

**Revised Figure 1 with adjusted y-axis.** To present a more objective assessment of our performance gains, we have redrawn Figure 1 with the y-axis starting at zero. As shown in Figure 8, the gains achieved by our method remain clearly visible.

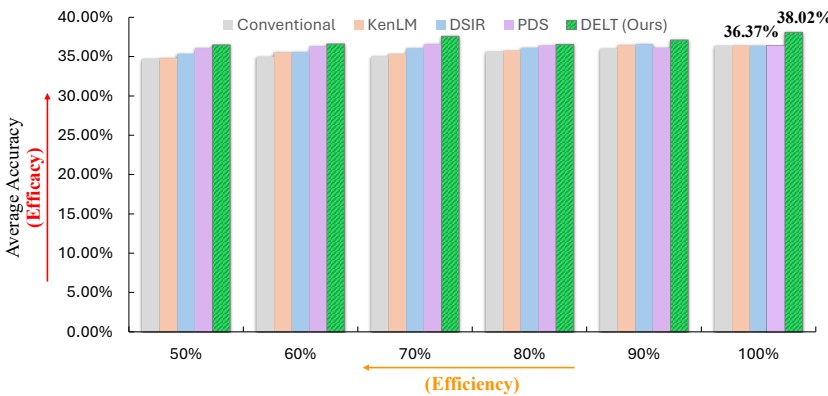

Figure 8: Redrawn Figure 1 with the y-axis starting at zero. Average result across 8 benchmarks for different methods.

## F.2 FOLDING LEARNING ANALYSIS

### F.2.1 DATA BIAS ANALYSIS

Traditional curriculum learning restricts the model to only "easy sample subsets" during early training (Bengio et al., 2009). This high homogeneity may restrict the gradient descent optimization path, causing the model to converge to a difficult-to-escape sharp local minimum (Soviany et al., 2022). To investigate the impact of the FO on data bias, we evaluate its advantages from two perspectives: *semantic diversity* and *difficulty diversity*. Specifically, we define a sliding window of size $w$ (equal to the batch size) and analyze the data within each window.

For semantic diversity, we compute the average pairwise distance of sample features within the window. These features are represented by the penultimate layer embeddings of GPT-3 and the distance is measured using the Euclidean metric. For difficulty diversity, we compute the standard deviation of the scores within the window.

As shown in Table 15, compared to the sorting method, the folding strategy significantly enhances the local diversity of the data. This improvement prevents the model from encountering performance bottlenecks caused by overly homogeneous samples within local regions.

Table 15: Effect of the fold layer.

| Methods | Semantic | Difficulty |
|---------|----------|-----------|
| Sorting | 0.060 | 0.0003 |
| **Folding** | 0.067 | 0.0014 |

### F.2.2 MODEL FORGETTING ANALYSIS

Traditional CL starts training from the easiest data bucket (Bengio et al., 2009). This sequential curriculum causes the model to lose contact with previously learned data, leading to the catastrophic forgetting of even simple concepts learned early on (Wang et al., 2021).

To directly monitor the forgetting phenomenon, we designed a layered validation experiment. We track the Perplexity (PPL) on the easiest 10% of the dataset ($D_{val\_easy}$) throughout training. As shown in Figure 9, the PPL of traditional CL (sorting) on $D_{val\_easy}$ rapidly drops (initial 30% phase) but then significantly rebounds after entering the "hard" data region (latter 50%), which directly confirms the forgetting of simple samples. For the FO (L=3), PPL drops normally in Fold 1. When simple data is reintroduced in Fold 2, PPL shows a secondary sharp drop (Re-learning). By

the end of training (Fold 3), the model maintains an extremely low PPL on $D_{val\_easy}$ and exhibits no rebound phenomenon seen in CL.

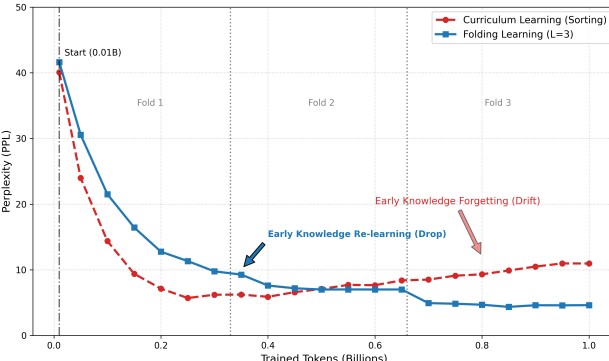

Figure 9: The LMs' perplexity (PPL) for $D_{val\_easy}$. Results obtained from a 160M Mistral model trained on 1B-token data (Evaluated every 0.1B tokens).

## G  LQS ANALYSIS

### G.1  VISUALIZATION OF SAMPLE LEARNABILITY SCORES AND GRADIENTS

To empirically validate the relationship between learnability scores and sample characteristics, we visualize the gradient norm dynamics by tracking the gradient norms and learnability scores of the 5 samples across different training stages. Specifically, these samples consist of one hard and one simple high-quality sample (approx. 800–1000 tokens), two long noisy samples (approx. 800–1000 tokens), and one short noisy sample (approx. 30 tokens).

As shown in Figure 10, the learnability score for high-quality samples (driven by gradient reduction) consistently outweighs the variance observed in noisy samples, ensuring that learnable high-quality data is assigned significantly higher scores. Additionally, long noisy samples exhibit a slight decline, the magnitude of which likely correlates with their specific noise levels. In contrast, the gradient norm of the short noisy sample stabilizes at a very early stage, suggesting that it has likely been memorized (overfitted) by the model.

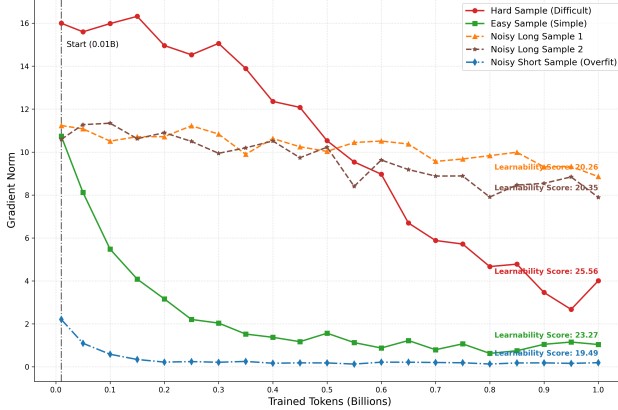

Figure 10: The gradient normalization results for 5 different data sample. Results obtained from a 160M Mistral model trained on 1B-token data (Evaluated every 0.1B tokens).

## G.2 SAMPLES SCORED BY LQS

To better illustrate our advantages, we visualized part samples. Specifically, we randomly sampled examples from the top 10% and bottom 10% based on their scores for visualization.

As shown in Table 16 and 17, high-scoring samples are high-quality, complex sentences that are not only challenging but also highly learnable, significantly aiding the model's optimization in later stages. In contrast, as shown in Table 18, low-scoring samples are often noisy and provide little to no benefit for model training, making them ideal candidates for filtering (see Table 18 example 1 and 2). However, some low-scoring samples consist of simple words or short phrases (see Table 18 example 3 and 4), which are beneficial for the model's learning in the early stages.

## Examples of Randomly Sampled High-Scoring Data Points (Top 10%)

# Example 1

 The precise nature of his doubt is not articulated but he aligns himself with the 17th century dissenters who put personal religious conscience before the unifying demands of the Church. This precipitates the move to the North, and determines the whole narrative arc of the novel. Before we are thrust, along with our central characters, into the smoky North, it is worth lingering a while on this short first stage of the novel. Gaskell does not make a simple thing of the South, as she might have been tempted to do as a clear point of comparison. Instead she offers us two versions of the South, two kinds of imaginings, both of which are then rejected. The drawing-room world of the Shaws, while superficially appealing, is altogether too enervating for the Margaret Hale who is gradually emerging even in these early chapters. Her decided refusal of Lennox is also a refusal of that world. The rural delights of Helstone (in the New Forest) seem initially to offer a simpler, perhaps a truer, version of the South. But that has already been put in doubt by Margaret and Henry Lennox's rather vexed discussion of it. In London, Henry suggests playfully that it is a 'village in a tale', at which Margaret takes umbrage, only to offer instead that it is 'a village in a poem' (11). When Henry arrives bang in the middle of that poem, the scene is set for romance: 'velvety cramoisy roses' (25), pears plucked from the tree and arranged on a plate of beetroot leaf, and the 'crimson and amber foliage' (26) of the deep forest beyond. Yet instead of the completion of the romantic dream, he comes up hard against Margaret's refusal. Indeed she herself comes up hard against it, and looks back at his proposal somewhat wistfully when she is plunged into her father's ferment, and briefly longs for the London/Shaw world where nothing 'called for much decision'. But if Margaret had accepted Henry { no novel. And besides, he can be kept in mind as a possible future plot line. Gaskell is astute enough to know that challenge makes for a more interesting narrative, and as it turns out, decision is something Margaret is rather good at. In the move to the South, she becomes the adult of the household, her mother declining into frailty, her father exhausted by the consequences of his own conscience. Her growth into her own strength of being is the more convincing because she often quails at what is before her. 'But the future must be met, however stern and iron it be.' (55) Thus, after a brief lull and taking of rest at the seaside town of Heston, Margaret and her father make the journey to Milton { the 'North' of the novel { where, as she says playfully, '\I am overpowered by the discovery of my own genius for management."' (57) But the obstacles are real, and the whole family must contend with their much-changed situation: They were settled in Milton, and must endure smoke and fogs for a season; indeed all other life seemed shut out from them by as thick a fog of circumstance. ... At night when Margaret realised this, she felt inclined to sit down in a stupor of despair. The heavy smoky air hung about her bedroom, which occupied the long narrow projection at the back of the house. The window, placed at the side of the oblong, looked to the blank wall of a similar projection, not above ten feet distant. It loomed through the fog like a great barrier to hope. (62) We are almost in the world of George Orwell's The Road to Wigan Pier { a work in fact heavily influenced by nineteenth-century depictions of urban industrialised living conditions. As if to underline their changed life, a letter has come from Edith, full of the delights of married life in Corfu: 'Edith's life seemed like the deep vault of blue sky above her, free { utterly free from fleck or cloud'. (62) This leads Margaret to reflect in turn on how, if she had accepted Lennox's marriage proposal, things might have been different. The omniscient narrative is here able to give much insight into Margaret's inner thoughts, so that we see her working through these difficult ideas and eventually finding herself clearer and happier: 'As she realised what might have been, she grew to be thankful for what was.' (63) If there is a measure of rationalisation in Margaret's logic here { that too is realistic.

---

# Example 2

 For example, signals 116 and 118 may be in-phase (I) and quadrature (Q) baseband components of a signal. In the example of FIG. 1B, signals 116 and 118 undergo a zero crossing as they transition from +1 to -1. Signals 116 and 118 are multiplied by signal 120 or signal 120 phase shifted by 90 degrees. Signal 116 is multiplied by a 0 degree shifted version of signal 120. Signal 118 is multiplied by a 90 degree shifted version of signal 120. Resulting signals 122 and 124 represent time-varying complex carrier signals. Note that signals 122 and 124 have envelopes that vary according to the time-varying amplitudes of signals 116 and 118. Further, signals 122 and 124 both undergo phase reversals at the zero crossings of signals 116 and 118. Signals 122 and 124 are summed to result in signal 126. Signal 126 represents a time-varying complex signal. Signal 126 may represent an example input signal into VPA embodiments of the present invention. Additionally, signals 116 and 118 may represent example input signals into VPA embodiments of the present invention. 1.2) Example Generation of Time-Varying Complex Envelope Signals from Constant Envelope Signals The description in this section generally relates to the operation of step 508 in FIG. 50. FIG. 1C illustrates three examples for the generation of time-varying complex signals from the sum of two or more substantially constant envelope signals. A person skilled in the art will appreciate, however, based on the teachings provided herein that the concepts illustrated in the examples of FIG. 1C can be similarly extended to the case of more than two constant envelope signals. In example 1 of FIG. 1C, constant envelope signals 132 and 134 are input into phase controller 130. Phase controller 130 manipulates phase components of signals 132 and 134 to generate signals 136 and 138, respectively. Signals 136 and 138 represent substantially constant envelope signals, and are summed to generate signal 140. The phasor representation in FIG. 1C, associated with example 1 illustrates signals 136 and 138 as phasors P136 and P138, respectively. Signal 140 is illustrated as phasor P140. In example 1, P136 and P138 are symmetrically phase shifted by an angle $\phi_1$ relative to a reference signal assumed to be aligned with the real axis of the phasor representation. Correspondingly, time domain signals 136 and 138 are phase shifted in equal amounts but opposite directions relative to the reference signal. Accordingly, P140, which is the sum of P136 and P138, is in-phase with the reference signal. In example 2 of FIG. 1C, substantially constant envelope signals 132 and 134 are input into phase controller 130. Phase controller 130 manipulates phase components of signals 132 and 134 to generate signals 142 and 144, respectively. Signals 142 and 144 are substantially constant envelope signals, and are summed to generate signal 150. The phasor representation associated with example 2 illustrates signals 142 and 144 as phasors P142 and P144, respectively. Signal 150 is illustrated as phasor P150. In example 2, P142 and P144 are symmetrically phase shifted relative to a reference signal. Accordingly, similar to P140, P150 is also in-phase with the reference signal. P142 and P144, however, are phase shifted by an angle whereby $\phi_2 \neq \phi_1$ relative to the reference signal. P150, as a result, has a different magnitude than P140 of example 1.

Table 16: Examples of Randomly Sampled High-Scoring Data Points (Top 10%).

## Examples of Randomly Sampled High-Scoring Data Points (Top 10%)

# Example 3

```
 Then Eddie owned up and said I took the records and my mummy said what did you do with them Eddie
and he said I played cards with them that's what I done with them where as everybody was playing
cards for tuppence and thrupence and he was playing with the records and our Tommy stood and looked
at him, I have never forgotten the expression on Tommy' s face.  Eddie was about 15 or 16 then.
When Eddie left school he successfully applied for a job in the Belfast City Council and I remember
everybody being very proud because it was difficult to get a job with the Council back then.  My
mummy came in one day and said I was talking to the foreman about our Eddie and he said he's a great
worker, my mummy was very proud of him.  He used to land in for his lunch to my mummy with all the
other binmen, she would have to feed them all.  He stayed there until 1968 when he began working
at boarding up buildings that had been damaged in the Troubles.  As a way to earn an extra bit of
money for the family he also worked nights as a barman.  When Eddie got older he was always very
particular about his appearance, he always wore a suit, sometimes with shirt or tee shirt, he was
always very spick and span.  Eddie smoked but he wasn't a drinker.  That's not to say that he didn't
try it at the beginning but it wasn't for him, he became a lifelong pioneer and a blood donor.  He
was also in the Confraternity (which was a sort of prayer group for men) at Clonard Monastery and
he loved it; that was his wee place to get away to.  He had strong faith.  When Eddie was sixteen
he met and fell in love with his future wife Marie.  Marie was the love of his life and they courted
for six years before getting married in 1962.  Five years later their first child was born, quickly
followed by three more.  Eddie and Marie had three sons { Eamon, Patrick and Ciaran, and one daughter
{ Brenda.  When they got married they went to live with Marie's grandmother in Fort Street.  But he
wanted his own house for himself and Marie and the only way that was going to happen was to get the
money together to buy one.  My daddy said to him.  \look if you're looking extra money to buy a house
go and join the TA its only 2 months a year".  So he went and joined the Territorial Army and the
money he was getting he sent it home to Marie.  He wasn't in the TA for very long and had left by the
time his first child was born.  When Eddie came home things didn't work out the way he wanted about
the house and things got too much for him, he ended up with a bit of a breakdown.  Eventually they
got the money for a house in Iveagh Street it was in a bad state of repair but Eddie and Marie fixed
it up and made it their home.  He suffered with mental health difficulties a couple of times in the
early-mid 1960s but that was well behind him by the time of his death.  Eddie just lived for Marie
and their kids, he took on a couple of extra jobs, working as a bar man and doing a bit of painting
and decorating.  He was always ready and willing to drop everything and do something for you.  It
was just an ordinary family life and he just loved Marie, he idolised her and she could do no wrong
in his eyes.  All he had time for was work, home and the confraternity.  When he did have free time
he liked fishing and clay pigeon shooting.  He was content with what he had and he was in his own
wee orbit that he owned his house, and provided for his kids he was just happy to be a husband and a
daddy.  On payday he would give Marie his unopened pay packet, she would then buy him his cigarettes
for the week.  Not too many men did that in those days.  Eddie was strict in a way too with the kids,
I remember Eddie coming to visit me with Ciaran, I had a rocking horse in the living room and Ciaran
wanted on it and Eddie said no you're not going over it doesn't belong to you, and I looked at Eddie
and said let the child go over and get on to the horse I said catch yourself on there is nobody even
on it and he went over but he was holding him on it because he maybe would of toppled.
```

# Example 4

```
 I was interested to see if I would lean closer to earlier poems or later poems since sometimes
there can be a significant difference in a poets writing style compared to when they began and
ended.  Turns ou This.Was.My.Jam Where do I even begin?  So the collection is written in reverse
chronological- yeah that's right I actually read the introduction to something.  I found this
particularly interesting because I feel like we often start in the beginning and naturally work our
way through their work.  I was interested to see if I would lean closer to earlier poems or later
poems since sometimes there can be a significant difference in a poets writing style compared to
when they began and ended.  Turns out I pretty steadily loved it all.  I think if I HAD to chose
I would lean just slightly closer to the beginning of the collection, but just slightly.  That
might have a bit of a biases though since Annabelle Leigh is the very first poem we read and it's
always been my absolute favorite.  Annabelle Leigh aside, I can only imagine what other wonderfully
powerful and hauntingly beautiful pieces he could have continued to write had he lived longer.
(Internally sobbing) You'll probably notice that there's a lot of reoccurrence with things like the
moon, celestial bodies, night, and the evening star- all things I really enjoyed.  Also, (and this
might quite well be my favorite) Poe has some of the best rhymes.  Words that rhymed but weren't your
usual rhymes, if you will.  For example:  departed and brokenhearted, month of June and mystic-moon,
dipt in folly and melancholy, Heaven and unforgiven (you gotta twang a little for that one), itself
alone and gray tombstone, heart's content and own element.  etc.  And it doesn't stop there!  The
the entire language being used is SO GOOD. I'd be reading a poem and then I'd hit a particular
line or phrase and just have to a take a moment to say "damn" while the words were absorbed.  Some
examples of that are "And the silken, sad, uncertain rustling of each purple curtain" (The Raven),
I stand amid the roar of a surf-tormented shore (A Dream Within a Dream), With the moon-tints of
purple and pearl (Eulalie-A Song), Sound loves to revel in a summer night:  Witness the murmur of
the gray twilight.  (Al Aaraaf Part 2) and "So like you gather in your breath, a portrait taken after
death.  (Tamberlane) Even the poems that I didn't mark as favorites I still really enjoyed.  My least
favorite in the collect was Al Aaraaf (both parts), I'm not really sure why I just didn't feel as
wow'ed by it.  Also, the play that ends the collection I wasn't a huge fan of but I think that just
speaks true to the format.  Plays are different than poetry.  I haven't read any Poe stories for a
long time, so I think it would be interesting to see where my enjoyment falls on the prose.  But,
the poetry is definitely out of the park for me.  Something I do find intriguing is that growing up
I also thought Poe was just a dark and haunted poet.  I think he was in fact haunted, but I don't
think (the poetry at least) is as horrific as people usually indicate.  In fact, I'm willing to
call it beautiful.  Beautifully dark, perhaps?  Read it, it's perfect.  Mateo { Oct 24, 2020 I did
not make this image but this is my review I did not make this image but this is my review Stephanie
Grosse { Sep 23, 2018 This review has been hidden because it contains spoilers.  To view it, click
here.  Simultaneously mysterious and familiar, like the old friend who suddenly astonishes you with
his strangeness or the acquaintance whom you are convinced you must have known since childhood.  I
very much enjoyed the use of onomatopoeia.  You will be hypnotised by the sounds (for example "ee",
"em" in the summer dream beneath the tamarind tree).  Poe has you forever, in "a dream within a
dream" Very memorable.A must read for all poetry lovers.  Simultaneously mysterious and familiar,
like the old friend who suddenly astonishes you with his strangeness or the acquaintance whom you are
convinced you must have known since childhood.  I very much enjoyed the use of onomatopoeia.
```

27

Table 17: Examples of Randomly Sampled High-Scoring Data Points (Top 10%).

---

### Examples of Randomly Sampled Low-Scoring Data Points (Bottom 10%)

# Example 1

```
XXXXXXXXXXXXXXXXXXXXXXXXXXXXXXXXX XXXXXXXXXXXXXXXXXXXXXXXXXXXXXXXXXXXXXXXXXXXX
XXXXXXXXXXXXXXXXXXXXXXXXXXXXXXXXXXXXXXXXXXXXXXXXXXXXXXXXXXXXXXXXXXXXXXXXXXXXXXXXXXXX
XXXXXXXXXXXXXXXXXXXXXXXXXXXXXXXXXXXXXXXXXXXXXXXXXXXXXXXXXXXXXXXXXXXXXXXXXXXXXXXXXXXX
XXXXXXXXXXXXXXXXXXXXXXXXXXXXXXXXXXXXXXXXXXXXXXXXXXXXXXXXXXXXXXXXXXXXXXXXXXXXXXXXXXXX
XXXXXXXXXXXXXXXXXXXXXXXXXXXXXXXXXXXXXXXXXXXXXXXXXXXXXXXXXXXXXXXXXXXXXXXXXXXXXXXXXXXX
XXXXXXXXXXXXXXXXXXXXXXXXXXXXXXXXXXXXXXXXXXXXXXXXXXXXXXXXXXXXXXXXXXXXXXXXXXXXXXXXXXXX
XXXXXXXXXXXXXXXXXXXXXXXXXXXXXXXXXXXXXXXXXXXXXXXXXXXXXXXXXXXXXXXXXXXXXXXXXXXXXXXXXXXX
XXXXXXXXXXXXXXXXXXXXXXXXXXXXXXXXXXXXXXXXXXXXXXXXXXXXXXXXXXXXXXXXXXXXXXXXXXXXXXXXXXXX
XXXXXXXXXXXXXXXXXXXXXXXXXXXXXXXXXXXXXXXXXXXXXXXXXXXXXXXXXXXXXXXXXXXXXXXXXXXXXXXXXXXX
XXXXXXXXXXXXXXXXXXXXXXXXXXXXXXXXXXXXXXXXXXXXXXXXXXXXXXXXXXXXXXXXXXXXXXXXXXXXXXXXXXXX
XXXXXXXXXXXXXXXXXXXXXXXXXXXXXXXXXXXXXXXXXXXXXXXXXXXXXXXXXXXXXXXXXXXXXXXXXXXXXXXXXXXX
XXXXXXXXXXXXXXXXXXXXXXXXXXXXXXXXXXXXXXXXXXXXXXXXXXXXXXXXXXXXXXXXXXXXXXXXXXXXXXXXXXXX
Select a country but NOT a region All you need to know about the Indian Defence Forces!
```

---

# Example 2

```
XXXXXXXXXXXXXXXXXXXXXXXXXXXXXXX XXXXXXXXXXXXXXXXXXXXXXXXXXXXXXXXXXXXXXXXX
XXXXXXXXXXXXXXXXXXXXXXXXXXXXXXXXXXXXXXXXXXXXXXXXXXXXXXXXXXXXXXXXXXXXXXXXXXXXXXXXXXXX
XXXXXXXXXXXXXXXXXXXXXXXXXXXXXXXXXXXXXXXXXXXXXXXXXXXXXXXXXXXXXXXXXXXXXXXXXXXXXXXXXXXX
XXXXXXXXXXXXXXXXXXXXXXXXXXXXXXXXXXXXXXXXXXXXXXXXXXXXXXXXXXXXXXXXXXXXXXXXXXXXXXXXXXXX
XXXXXXXXXXXXXXXXXXXXXXXXXXXXXXXXXXXXXXXXXXXXXXXXXXXXXXXXXXXXXXXXXXXXXXXXXXXXXXXXXXXX
XXXXXXXXXXXXXXXXXXXXXXXXXXXXXXXXXXXXXXXXXXXXXXXXXXXXXXXXXXXXXXXXXXXXXXXXXXXXXXXXXXXX
XXXXXXXXXXXXXXXXXXXXXXXXXXXXXXXXXXXXXXXXXXXXXXXXXXXXXXXXXXXXXXXXXXXXXXXXXXXXXXXXXXXX
XXXXXXXXXXXXXXXXXXXXXXXXXXXXXXXXXXXXXXXXXXXXXXXXXXXXXXXXXXXXXXXXXXXXXXXXXXXXXXXXXXXX
XXXXXXXXXXXXXXXXXXXXXXXXXXXXXXXXXXXXXXXXXXXXXXXXXXXXXXXXXXXXXXXXXXXXXXXXXXXXXXXXXXXX
XXXXXXXXXXXXXXXXXXXXXXXXXXXXXXXXXXXXXXXXXXXXXXXXXXXXXXXXXXXXXXXXXXXXXXXXXXXXXXXXXXXX
XXXXXXXXXXXXXXXXXXXXXXXXXXXXXXXXXXXXXXXXXXXXXXXXXXXXXXXXXXXXXXXXXXXXXXXXXXXXXXXXXXXX
XXXXXXXXXXXXXXXXXXXXXXXXXXXXXXXXXXXXXXXXXXXXXXXXXXXXXXXXXXXXXXXXXXXXXXXXXXXXXXXXXXXX
Select a country but NOT a region A site focusing on Australian Modelling with galleries, articles
and discussion forums.
```

---

# Example 3

```
USA, Liberia USA, Lithuania, Italy USA, Luxembourg USA, Luxembourg, UK USA, Malaysia USA, Malta,
France, UK USA, Malta, UK USA, Mexico USA, Mexico, Australia USA, Mexico, Australia, Canada USA,
Mexico, Canada USA, Canada, Germany USA, Mexico, Germany USA, Mexico, Hong Kong USA, Mexico,
Japan USA, Mexico, Spain USA, Mexico, UK USA, Mexico, United Arab Emirates USA, Monaco USA, Monaco,
Morocco USA, Morocco USA, Morocco, Spain, UK USA, Morocco, Switzerland USA, Myanmar USA, Netherlands
USA, Netherlands, France USA, Netherlands, Germany, France, Austria USA, Netherlands, South Africa
USA, Netherlands, UK USA, Netherlands, UK, Denmark USA, New Zealand USA, New Zealand, Canada, Israel,
Japan, Nigeria USA, New Zealand, Germany USA, New Zealand, Japan USA, New Zealand, South Africa, UK,
Lithuania USA, New Zealand, UK USA, Nicaragua USA, Nigeria USA, Norway USA, Pakistan USA, Panama,
Argentina USA, Panama, Japan, Canada USA, Panama, Mexico USA, Peru USA, Philippines USA, Philippines,
Puerto Rico USA, Philippines, Taiwan, South Korea, China, Canada USA, Poland USA, Poland, Slovenia,
Czech Republic, UK USA, Portugal USA, Portugal, France USA, Puerto Rico USA, Qatar USA, Romania USA,
Romania, Canada USA, Romania, France, Italy, Germany USA, Romania, Germany USA, Romania, Iceland USA,
Romania, UK USA, Russia USA, Russia, Hungary USA, Russia, UK USA, Saudi Arabia USA, Senegal USA,
Serbia USA, Serbia, Canada USA, Singapore USA, Singapore, Taiwan USA, Slovakia USA, Slovakia, China
USA, South Africa USA, South Africa, Germany USA, South Africa, India USA, South Africa, Italy USA,
South Africa, Zambia, Germany USA, South Korea USA, South Korea, Australia USA, South Korea, India
USA, South Korea, Japan USA, South Korea, Singapore USA, South Korea, Singapore, Russia, Malaysia,
Kazakhstan, Taiwan, Hong Kong, Japan, China, India, Syria, Iran, Egypt, Pakistan USA, South Korea,
Spain
```

---

# Example 4

```
Phillip L. Horrell v. David Gomez, Warden, No. 20-5306 Ganaa Otgoo v. Illinois, No. 20-5109
Phillip Hartsfield v. Stepanie Dorethy, Warden, No. 19-1473 Anthony Jackson v. Supreme Court
of Illinois, No. 19-8665 David Beverly v. Illinois, No. 19-8502 Lamont Dantzler v. Illinois,
No. 19-8448 Joh-ner Taylor Wilson v. Illinois, No. 19-8437 Herbert Burgess v. Illinois, No.
19-8379 Joseph M. Coffman v. Illinois, No. 19-8391 Timothy J. McVay v. Illinois, No. 19-8304
Kenneth Durant v. Frank Lawrence, Warden, No. 19-7967 Seth A. Weaver v. Illinois, No. 19-7823
Lyarron T. Emers v. Illinois, No. 19-7759 Bethany Austin v. Illinois, No. 19-1029 Anthony Allen
v. Illinois, No. 19-7633 Kenin L. Edwards v. Michael L. Atterberry, et al., No. 19-965 Pablo
Rodriguez-Palomino v. Illinois, No. 19-7273 Christopher L. Croom v. Illinois, No. 19-7237 Tony
Robinson v. Illinois, No. 19-7226 Lazaro Zapata v. Illinois, No. 19-7264 Fernando Oliveros v.
Illinois, No. 19-7141 Kevin Dameron v. Illinois, No. 19-6945 Peter Gakuba v. Michelle Neese, No.
19-6543 Richard Kalinowski v. Illinois, No. 19-6368 Rafael Alvarado v. Frank Lawrence, Warden,
No. 19-6347 Chad M. Cutler v. Illinois, No. 19-6150 Hezekiah Whitfield v. Deanna Brookhart,
Warden, No. 19-6051 Lorenzo Davis, Jr. v. Illinois, No. 19-5831 Chadwick N. Barner v. Illinois,
No. 19-5655 Charles Donelson v. Q. Tanner, et al., No. 19-5397 Robert Curry v. Illinois, No.
19-5366 Andrew Condon v. Illinois, No. 19-5349 Keith Talbert v. Illinois, No. 18-9768 Juan
Rodriguez v. Illinois, No. 18-9759 Miguel Alcantar v. Illinois, No. 18-1548 Gregory Rayford
v. Illinois, No. 18-9612 Irving Madden v. Michael Melvin, Warden, No. 18-9474 Denzel Pittman v.
Illinois, No. 18-9451 Jesus Cotto v. Jacqueline Lashbrook, Warden, No. 18-9116 Pierre Montanez v.
Ursula Walowski, No. 18-9101 Russell Frey v. Illinois, No. 18-9120 Peter Gakuba v. Illinois, No.
18-9041 Jose Cobian v. Illinois, No. 18-8963 Derrick Redmond v. Illinois, No. 18-8808 Jennifer N.
Nere v. Illinois, No. 18-8625 Gerald W. Long v. Illinois, No. 18-8577 Willie White v.
```

---

Table 18: Examples of Randomly Sampled Low-Scoring Data Points (Bottom 10%).

