# OpenReview forum: "Data Efficacy for Language Model Training"
_ICLR.cc/2026/Conference — Submitted to ICLR 2026_

### Official Review · Reviewer_dGm4 · 2025-10-20

**Soundness:** 2
**Presentation:** 2
**Contribution:** 2
**Rating:** 2
**Confidence:** 4

**Summary:**

This paper introduces a two-step framework, DLET, designed to improve data efficiency in LLM pre-training. It first scores a small subset of the training data, then trains a scoring model (similar to the reward model in RLHF), and finally employs this model to sort the training data using a novel folding approach.

The first contribution is the proposed scoring function, LQS, which evaluates the reliability, quality, and learnability of each training sample $x_n$ as $\gamma_n = \mathbf{R}(\boldsymbol{\theta}_{t+1}) \cdot \mathbf{Q}(x_n) \cdot \mathbf{L}(x_n)$.
The reliability function $R$ corresponds to the magnitude of the target vector $\lambda$ introduced by PDS. The quality function $Q$ measures the cosine similarity between the target vector $\lambda$ and the current gradient. Finally, the learnability function $L$ is defined as the cumulative fraction of gradients between the current and next steps.

The second contribution is the folding learning strategy, in which training samples are randomly divided, typically into three parts, and within each part, the data are greedily sorted according to their assigned scores.

**Strengths:**

The experiments are conducted in a comprehensive and convincing manner. Specifically, the comparisons cover three model scales, 160M, 470M, and 1B parameters, as well as strong baselines, such as PDS. The evaluation includes both pre-training and post-training results on the code and math domains. The ablation studies are also appropriately designed, focusing on (1) the folding learning strategy and (2) the proposed scoring function. Overall, I find the experimental evaluation to be thorough and persuasive.

**Weaknesses:**

**W1** The organization of the paper is somewhat confusing to me.

(1) The proposed scoring function, **LQS**, is composed of three key elements. However, only two of them—the quality and learnability functions—are described in the main body, while the reliability function is deferred to the appendix (page 16). It is unclear why such an important component is omitted from the main discussion, and this may lead to misunderstandings about the formulation of the proposed scoring function (e.g., Eq. 5).

(2) I personally find that **Section 4, “DELT Paradigm,”** adds limited clarity or new insight. In contrast, some critical implementation details are missing from the main body, such as the fact that the scoring model is trained on a small subset of the training data, and that most score annotations are generated by this model’s predictions.

**W2** The **motivation for the folding learning** component also appears weak. While I appreciate the comprehensive experimental evaluation that supports its performance gains, the paper’s motivation for this technique is rather superficial. The single-sentence justification, that “they often face challenges such as data distribution bias and model forgetting”, does not sufficiently explain the conceptual reasoning behind folding learning. Moreover, I find the evidence presented in Table 14 (page 21), which claims that folding enhances data diversity, unconvincing.

**W3** The novelty of the proposed quality and reliability functions seems limited. The quality function is defined as the (cumulative) cosine distance between the target vector and the current gradient, while the reliability function corresponds to the norm of the target vector. Since the target vector itself is directly borrowed from the well-known PDS paper, it is difficult to regard these components as novel contributions.

**Questions:**

I do not have any questions at this stage. Although the organization of the paper is somewhat weak, it is clear that the authors have presented the technical details and experimental results thoroughly.

---

> ### Author Response · Authors · 2025-11-24
> **Rebuttal 1: W1**
>
> Thank you for your constructive comments. We are greatly encouraged by your appreciation of our work as "comprehensive", "convincing", "thorough", and "persuasive".
>
> You also raised some points of weakness and questions, which we found very enlightening. We thoroughly analyzed the issues you mentioned and provided the following responses.
>
> > (W1) The proposed scoring function, LQS, is composed of three key elements. However, only two of them—the quality and learnability functions—are described in the main body, while the reliability function is deferred to the appendix (page 16). It is unclear why such an important component is omitted from the main discussion, and this may lead to misunderstandings about the formulation of the proposed scoring function (e.g., Eq. 5).
> I personally find that Section 4, “DELT Paradigm,” adds limited clarity or new insight. In contrast, some critical implementation details are missing from the main body, such as the fact that the scoring model is trained on a small subset of the training data, and that most score annotations are generated by this model’s predictions.
>
> We appreciate your constructive suggestions regarding the organization of our paper. We apologize for the confusion caused by the presentation due to limited page. After carefully considering your feedback, we have restructured the manuscript to prioritize the core method and implementation details.
>
> 1. **Structural revisions.** All changes are highlighted in ${\color{blue}\text{blue}}$  in the revision.
>
>     First, we move the derivation and analysis of the Reliability function from the Appendix D to the main text (${\color{blue}\text{Section 3.2}}$). This ensures a coherent and complete presentation of the LQS formulation (Eq. 5).
>
>     Secondly, we relocate implementation details (including proxy data sampling, proxy data annotation, data scorer training, and full data scoring) from the Appendix E to the main text (${\color{blue}\text{Section 3.3}}$). This provides a clear, step-by-step view of the LQS pipeline.
>
>     Third, we move the conceptual discussion of the DELT paradigm to the (${\color{blue}\text{Appendix D}}$) to avoid distracting from the main contributions.
>
> 2. **Clarification on the DELT paradigm.** The DELT framework is defined as a system aiming to improve model performance by optimizing the utilization of training data. As presented in our paper, DELT currently encompasses data scoring and data ordering, while also supporting the integration of data selection.
>
>     Recent survey [1] related to data and language models indicates that a significant number of works focus on **altering data scales**, including data expansion methods (e.g., data acquisition, mixing and synthesis), as well as data reduction methods (e.g., deduplication, filtering and selection). In contrast, effective utilization at a **fixed data scale** (e.g., data ordering) has received far less attention. Not only is it not allocated a dedicated section, but it is typically represented by only 1 or 2 related papers. The DELT paradigm not only reignites interest in data ordering but also pioneers the efficient execution of simultaneous data selection and ordering using a unified data scoring metric. The key insight of the DELT paradigm lies in its systematic introduction of data efficacy, which brings structured attention to the optimization of data utilization at a fixed data scale.
>
>     In our future roadmap, the DELT framework will include additional modules beyond scoring and ordering to further enhance data efficacy. Therefore, the proposed LQS & FO methods are regarded as one specific instantiation of DELT. Furthermore, our full code implementation (provided in the supplementary material) is architected strictly according to the DELT paradigm. Our primary motivation for introducing this framework was to pave both the theoretical and practical way for our future work on data efficacy.
>
>     However, we understand your concern that the DELT framework may currently offer limited insight and could potentially distract readers from the main contributions of this paper. Consequently, we move the description of DELT to the ${\color{blue}\text{Appendix D}}$.
>
> [1] A Survey of LLM x DATA. Arxiv 2025.

---

> ### Author Response · Authors · 2025-11-24
> **Rebuttal 2: W2**
>
> > (W2) The motivation for the folding learning component also appears weak. While I appreciate the comprehensive experimental evaluation that supports its performance gains, the paper’s motivation for this technique is rather superficial. The single-sentence justification, that “they often face challenges such as data distribution bias and model forgetting”, does not sufficiently explain the conceptual reasoning behind folding learning. Moreover, I find the evidence presented in Table 14 (page 21), which claims that folding enhances data diversity, unconvincing.
> Thank you for your penetrating question. We fully understand your concerns regarding the empirical results of folding ordering. We apologize for not sufficiently explaining the confusion caused by model forgetting and data distribution bias associated with Curriculum Learning (CL) in the paper.
>
> In traditional CL, previous works [2,3] have already pointed out that model forgetting and data distribution bias are two widely recognized issues, which become severe in LLM training where only one or few epochs are applied.
>
> 1. **Model forgetting.** Traditional CL starts training from the easiest data bucket [4]. This sequential curriculum causes the model to lose contact with previously learned data, leading to the catastrophic forgetting of even simple concepts learned early on [2]. **Existing Solutions** address model forgetting by using auxiliary models to detect 'forgetting' signals [5,6], dynamically increasing the replaying probability of data soon to be forgotten [9]. However, utilizing auxiliary models introduces substantial computational overhead. Additionally, simple replay strategies are incompatible with the single-epoch setting of mainstream LM pre-training, making these approaches impractical.
>
> 2. **Optimization traps due to data distribution bias.** Traditional CL restricts the model to only "easy sample subsets" during early training. This high homogeneity may restrict the gradient descent optimization path, causing the model to converge to a difficult-to-escape sharp local minimum [3]. **Existing Solutions** often involve regularization or mixing strategies to enforce the selection of more diverse samples, thereby avoiding local optima specific to a single difficulty group [7,8]. However, these methods often require designing specialized diversity metrics tailored to specific datasets, rendering them impractical.
>
> To mitigate these recognized issues, we introduced FO, a simple yet effective strategy. Specifically, it degenerates to traditional CL at $L=1$ and becomes random shuffling at $L \to \infty$. When configured with an appropriate value, FO effectively balances the sequential learning capability of CL with the robustness of random shuffling against forgetting and distribution bias.
>
> We validate its superior performance over traditional CL in addressing these two problems through the following experimental evidence.
>
> 3. **Validation on model forgetting.** The sequential nature of CL (simple $\to$ complex) can lead to the irreversible overwriting of "simple" knowledge. By introducing $L$ folds, the model effectively simulates the process of repeating earlier knowledge (pseudo-replay), thereby preventing catastrophic knowledge forgetting.
>
>     To directly monitor the forgetting phenomenon, we designed a layered validation experiment. We track the Perplexity (PPL) on the easiest 10% of the validation set ($D_{easy}$) throughout training. As shown in ${\color{blue}\text{Appendix F.2.2}}$ of the revision, the PPL of traditional CL (sorting) on $D_{easy}$ rapidly drops (initial 30% phase) but then significantly rebounds after entering the "hard" data region (latter 50%), which directly confirms the forgetting of simple samples. For the FO (L=3), PPL drops normally in Fold 1. When simple data is reintroduced in Fold 2, PPL shows a secondary sharp drop (Re-learning). By the end of training (Fold 3), the model maintains an extremely low PPL on $D_{easy}$ and exhibits no rebound phenomenon seen in CL.
>
> 4. **Validation on optimization traps.** The strict ordering in CL leads to high difficulty homogeneity within training batches. Theoretically, FO resolves this by folding, which intentionally reintroduces samples of differing difficulty within each micro-training window, restoring local data diversity and acting as an implicit regularizer.
>
>     We quantitatively analyzed the training data flow's difficulty diversity (measured by the standard deviation of sample scores within a sliding window, see ${\color{blue}\text{Appendix F.2.1}}$). The results show that traditional CL shows low difficulty diversity, confirming its severe distribution bias. In contrast, FO boosts this diversity (an increase of ~4.7x). This indicates that FO mitigates the local data distribution bias inherent in CL, helping the optimizer converge on a smoother and more representative loss landscape.

---

> ### Author Response · Authors · 2025-11-24
> **Rebuttal 3: W3**
>
> > (W3) The novelty of the proposed quality and reliability functions seems limited. The quality function is defined as the (cumulative) cosine distance between the target vector and the current gradient, while the reliability function corresponds to the norm of the target vector. Since the target vector itself is directly borrowed from the well-known PDS paper, it is difficult to regard these components as novel contributions.
>
> Thank you for your question. We have elaborated on the distinctions between our work and PDS [10], which we hope provides a clear understanding of our contribution. While adopt the target vector definition from PDS [10], our strategies diverge significantly in motivation, scoring, and ordering methods.
>
> 1. **Motivation and task.** PDS focuses only on data selection (efficiency), which identifies a sparse subset to reduce training cost. In contrast, our work introduces the broader concept of data efficacy. We focus on maximizing the performance gain from a fixed or selected budget by **optimizing data utilization** (i.e., how and when to present data).
>
> 2. **Scoring strategy.** PDS scores data purely based on the inner product $\lambda^\top \nabla l$ (Quality). Ideally, this maximizes the Hamiltonian function under PMP. However, in non-convex optimization, this can favor samples with exploding gradients or high variance. Therefore, we propose Learnability-Quality Scoring (LQS), which divides the PDS quality term by a learnability term (the future gradient norm $||\nabla l_{next}||$). This acts as a "Stability Regularizer" applied to the PDS theory. It penalizes samples that are directionally correct but difficult to converge (high uncertainty), ensuring we prioritize data that is both high-quality and efficiently learnable by the current model state. This allows LQS to serve **both ordering and selection** tasks more robustly than the raw PDS score.
>
> 3. **Ordering strategy.** Since PDS is designed for selection, it does not address the order of data presentation. We propose **Folding Ordering**, a novel scheduling strategy based on our LQS. FO leverages the "Learnability" component of our score to construct a learning process that mitigates the distribution bias and forgetting issues common in traditional sorting-based curricula.
>
> In summary, while PDS provides the "compass" (target vector $\lambda$), our work designs a better "path" (LQS and Folding Ordering) to reach the destination efficiently and stably.
>
> [2] A Survey on Curriculum Learning. T-PAMI 2021.
>
> [3] Curriculum Learning: A Survey. IJCV 2022.
>
> [4] Curriculum learning. ICML 2009.
>
> [5] Teacher-student curriculum learning. TNNLS 2019.
>
> [6] Automated curriculum learning for neural networks. ICML 2017.
>
> [7] Minimax curriculum learning: Machine teaching with desirable difficulties and scheduled diversity. ICLR 2018.
>
> [8] LAMOL: LAnguage MOdeling for Lifelong Language Learning. ICLR 2020.
>
> [9] Experience replay for continual learning. NeurIPS 2019.
>
> [10] Data Selection via Optimal Control for Language Models. ICLR 2025.

---

> > ### Comment · Reviewer_dGm4 · 2025-11-27
> > **Reviewer feedback**
> >
> > Thank you for the revision and for the clarifications on motivation, novelty, and contribution. I have raised my score from 2 to 4. I begin by briefly recapping my earlier concerns, followed by more detailed feedback.
> >
> > **W1. Organization (Addressed)**
> >
> > Previously, of the three key elements (Q, L, R), only the quality and learnability functions (Q, L) were described in the main text, while the reliability function (R) appeared only in the appendix. This made it difficult to understand the motivation and role of R.
> >
> > In the revision, both the motivation and technical details of R are moved into Section 3. Appreciate!
> >
> > **W2. Motivation of folding learning (Partially addressed)**
> >
> > Previously, the motivation for folding learning relied mainly on the single sentence that curriculum learning “often faces challenges such as data distribution bias and model forgetting,” which I found too weak on its own.
> >
> > The revision now includes an empirical investigation (Figure 9, Appendix, p. 24) of the PPL evolution on the evaluation set under different sorting methods (CL). The observation that CL yields slightly higher PPL on easy evaluation samples suggests that CL may forget easy samples and thus potentially helps motivate folding learning.
> >
> > I find this direction promising and appreciate the added analysis. At the same time, I do not yet see this as closed-loop evidence. It remains unclear whether higher PPL on “easy samples” consistently translates into worse downstream performance, especially if those samples are noisy or low-quality. My point is not to dispute the result, but to caution that strong claims like “CL is not helpful or even harmful for LLMs” require a broader set of experiments and more systematic evidence.
> >
> > **W3. Novelty of the proposed LQS scoring function (Not addressed)**
> >
> > My earlier concern was that, among the three proposed scoring functions, two depend heavily on the PDS target vector.
> >
> > The relevant rebuttal appears under **Rebuttal 3: W3, Scoring strategy**. While it provides a long explanation, it remains mostly qualitative and does not make me believe the LQS is a novel scoring function.
> >
> > **Kind suggestion**
> >
> > I raised my score from 2 to 4. This work has strong empirical results and a solid ablation study, but I still find it difficult to give a clearly positive score.
> >
> > The core issue is that the paper proposes **two non-trivial and orthogonal** components: folding learning and the LQS scoring function. Yet, neither is supported with a closed-loop line of evidence. Folding learning would require a very strong claim (and broad evidence) that curriculum learning is harmful in LLM pre-training. Likewise, the LQS scoring function would benefit from clearer technical motivation and a concrete demonstration of its advantage over the PDS target vector (or the solution in PDS).

---

> > > ### Author Response · Authors · 2025-12-01
> > > **Response to your comment**
> > >
> > > We sincerely thank you for your response.
> > >
> > > We are encouraged by your willingness to raise the score and your continued recognition, specifically that you "find this direction promising and appreciate the added analysis", consider our methods "non-trivial", and commend the "strong empirical results and a solid ablation study".
> > >
> > > We have carefully noted the new concerns raised in your response and provide further clarifications below. We hope these points will further highlight the contributions and innovations of our work.
> > >
> > > 1. **Regarding Folding Ordering.** First, we appreciate your recognition of our newly added experiments. Furthermore, we wish to re-emphasize that data distribution bias and model forgetting are widely recognized challenges in Curriculum Learning (CL), rather than new findings introduced by us. In previous response Rebuttal 2: W2, we have also detailed their underlying causes and how existing methods attempt to address them.
> > >
> > >     Second, **regarding your impression that we implied "CL is not helpful or even harmful for LLMs", we wish to clarify a critical misunderstanding:**
> > >     - **We have never made such a claim in either the rebuttal or the paper.** On the contrary, we consistently acknowledge the effectiveness of traditional CL. This is also evidenced by our results, where CL consistently outperforms the random baseline in most experiments.
> > >     - While traditional CL yields performance gains, its potential is constrained by two widely recognized issues: model forgetting and data bias. Our theoretical explanations and PPL analysis on simple samples confirm these specific limitations. By mitigating them, FO unlocks further performance improvements.
> > >
> > > 2. **Regarding LQS Scoring.** You mentioned that our previous explanation appeared qualitative. However, we would like to highlight that our previous response provided a detailed comparison distinguishing our method from PDS in terms of both motivation and methodological principles, detailing how we developed LQS based on the  established theoretical foundations of PDS. Moreover, our paper provides many quantitative empirical evidence showing that LQS significantly outperforms PDS.
> > >
> > >     In summary, these points constitute a comprehensive and persuasive demonstration of our method's advancements and advantages over PDS across three dimensions: motivation, theoretical formulation, and empirical results.

---

### Official Review · Reviewer_RCEA · 2025-10-30

**Soundness:** 3
**Presentation:** 3
**Contribution:** 3
**Rating:** 6
**Confidence:** 4

**Summary:**

This paper focuses on data efficacy in LLM training—optimizing data utilization to boost performance, distinct from data efficiency . It proposes Folding Ordering (FO) to mitigate distribution bias and model forgetting of traditional curriculum learning by reorganizing sorted data at fixed intervals ; Learnability-Quality Scoring (LQS), the first method unifying ordering and selection, via gradient-based learnability and quality metrics ; and the DELT paradigm (integrating scoring, ordering, optional selection) to harness data value without altering data/model . Experiments across model sizes (160M–1.7B), data scales (1B–50B tokens), and domains (general NLP, math, code) show DELT outperforms baselines by up to 1.65% in average accuracy, excelling in efficacy and efficiency . It concludes data efficacy is a promising LLM training area .

**Strengths:**

It proposes innovative, practical methods: Folding Ordering  mitigates data distribution bias and model forgetting of traditional curriculum learning ; Learnability-Quality Scoring , the first method unifying data ordering and selection, uses gradient-based metrics for robust sample evaluation . The DELT paradigm integrates these with optional selection, enhancing efficacy/efficiency without altering data or model . Comprehensive experiments (160M–1.7B models, 1B–50B tokens, general NLP/math/code) show DELT outperforms baselines by 1.65% in average accuracy, proving broad robustness . These make it a impactful foundation for LLM training.

**Weaknesses:**

1. LQS relies on small, high-quality datasets to compute downstream loss for scoring.
2. It lacks validation on larger LMs (e.g., 10B+ parameters) or exascale datasets, leaving uncertainty about scalability in state-of-the-art LLM training.
3. The optimal folding layer L=3 for FO is empirically determined but lacks mechanistic explanation, increasing adoption friction for practitioners . These gaps restrict its generalizability.

**Questions:**

As shown in the Weakness.

---

> ### Author Response · Authors · 2025-11-24
> **Rebuttal 1: W1**
>
> Thank you for your valuable feedback. We are greatly encouraged by your praise for our work as "innovative", "practical", and "impactful foundation".
>
> The concerns and questions you pointed out have provided valuable insights. We have addressed the issues you mentioned and present our responses below.
>
> > (W1) LQS relies on small, high-quality datasets to compute downstream loss for scoring.
>
> Thank you for pointing this out. You are correct that in our LQS implementation, we utilized different small, high-quality datasets as target datasets to calculate the downstream loss for varying domains. We would like to clarify that the selection of target datasets is a common practice and does not impose a significant burden on our method, as explained in the following three points.
>
> 1. **Using target datasets is a general practice.** Given that downstream tasks vary significantly in difficulty distribution and domain, it is challenging to devise a single selection and ordering strategy that universally fits all scenarios. Consequently, using a target dataset to guide data selection is a widely adopted practice in this topic [1,2,3,4,5].
>
> 2. **The selection of target datasets is guided by established research.** The principles for selecting target datasets have been discussed in some previous work [4,5] (e.g., prioritizing high quality and diversity over large scale). Guided by these established principles, selecting an appropriate target dataset is relatively straightforward and easy to implement, and does not impose a significant burden on our method.
>
> 3. **External target datasets are NOT mandatory.** In scenarios where no external data is available, a random subset of the training data $D$ can serve as the target dataset for calculating LQS. However, as shown in Table C1, using a random subset from $D$ (without external datasets) yields limited performance gains compared to using a curated target set. Therefore, we opted for specific high-quality target datasets in our paper to get better performance.
>
> Table C1. Effect of using different downstream datasets to compute $J(\theta)$ (160M Mistral, 1B-token data). We report the average accuracy on the OLMo benchmarks.
>
> | Method | $J(\theta)$ | Acc. |
> | :- | :- | :- |
> | Conventional | - | 36.37 |
> | LQS | Subset of $D$ | 36.83 |
> | LQS | LIMA | 38.08 |
>
> [1] Data Selection for Language Models via Importance Resampling. NeurIPS 2023.
>
> [2] Prioritized training on points that are learnable, worth learning, and not yet learnt. ICML 2022.
>
> [3] Data Selection via Optimal Control for Language Models. ICLR 2025.
>
> [4] Less: selecting influential data for targeted instruction tuning. ICML 2024.
>
> [5] DsDm: Model-Aware Dataset Selection with Datamodels. 2024. ICML 2024.

---

> ### Author Response · Authors · 2025-11-24
> **Rebuttal 2: W2, W3**
>
> > (W2) It lacks validation on larger LMs (e.g., 10B+ parameters) or exascale datasets, leaving uncertainty about scalability in state-of-the-art LLM training.
>
> Thank you for your insightful question regarding the scalability of our method to the 10B parameter regime. We acknowledge the importance of verifying efficacy at larger scales. However, training a model exceeding 10B parameters would require approximately 35 days (over 800 GPU-hours on 8x A100s) for a single epoch on just 50B tokens, and over 140 days for a Chinchilla-optimal training run (approx. 200B tokens). Due to these high computational resource and time constraints within the rebuttal window, physically conducting this experiment is tough.
>
> To address this, we rigorously extrapolated our performance using the Chinchilla Scaling Law [6], grounded in robust empirical data from our 160M–1.7B experiments. Specifically, we fitted the Chinchilla scaling law $L(N, D) = E + \frac{A}{N^\alpha} + \frac{B}{D^\beta}$ to the test losses of our 160M, 470M, 1B, and 1.7B models. We utilized 40 data points (losses recorded every 2.5B tokens during 10B-token training on Redpajama corpus [7]) and minimized the Huber loss [8] to determine the constants. As shown in the Table C2, the improvements with our method (LQS&FO) persist in pre-training recent large LMs, such as GPT-3 [9] and Llama family [10,11,12].
>
> Table C2. Scaling Law-based test loss extrapolation. Predicted losses for configurations ($N, D$) equivalent to GPT-3 175B, Llama 6.7B, Llama 2 70B, and Llama 3.1 405B.
>
> | Model | $N$ | $D$ | Conventional | LQS&FO |
> | :--- | :--- | :--- | :--- | :--- |
> | GPT-3 | 175B | 300B | 3.876 | **3.865** |
> | Llama | 6.7B | 1.0T | 3.953 | **3.901** |
> | Llama 2 | 70B | 2.0T | 3.858 | **3.835** |
> | Llama 3.1 | 405B | 15T | 3.832 | **3.826** |
>
> [6] Training compute-optimal large language models. NeurIPS 2022.
>
> [7] RedPajama: an Open Dataset for Training Large Language Models. NeurIPS 2024.
>
> [8] Robust Estimation of a Location Parameter. Springer New York 1992.
>
> [9] Language models are few-shot learners. NeurIPS 2020.
>
> [10] Llama: Open and efficient foundation language models. Arxiv 2023.
>
> [11] Llama 2: Open foundation and fine-tuned chat models. Arxiv 2023.
>
> [12] The llama 3 herd of models. Arxiv 2024.
>
> > (W3) The optimal folding layer L=3 for FO is empirically determined but lacks mechanistic explanation, increasing adoption friction for practitioners . These gaps restrict its generalizability.
>
> Thanks for your constructive feedback. When $L$ is set to 1, the FO degenerates into curriculum learning, which suffers from model forgetting issues [13]. Conversely, as $L$ approaches infinity, the FO converges to random shuffling, which avoids the issue of model forgetting but results in suboptimal performance. Therefore, the superior performance of $L=3$ stems from its optimal trade-off between retaining prior knowledge and learning new high-quality knowledge. It is worth noting that in our paper, we used $L = 3$ as it achieved the best results in the 160M-model, 1B-tokens-data setting. For the sake of consistency, we adopted $L = 3$ throughout the paper.
>
> Furthermore, based on your suggestion, we explored the selection of the optimal $L$ value as the model and data scale increase. As shown in Table C3, the optimal $L$ value increases with data scale, indicating a positive correlation between the two. However, as the model scale increases, the optimal $L$ value remains nearly unchanged, suggesting that the $L$ value is largely insensitive to model scale. The exploration of $L$ values in Table C3 provides guidance for its selection, allowing practitioners to estimate an appropriate $L$ based on their data scale.
>
> Additionally, based on your insightful suggestion, we plan to further explore the scaling law between the $L$ value of folding learning, model scale, and data scale under larger models and datasets. Thank you for your valuable contribution to this work.
>
> Table C3: The optimal $L$ value varies with changes in model and data scale.
>
> | model/data | 1B-tokens | 10B-tokens | 30B-tokens | 50B-tokens |
> | ---------- | --------- | ---------- | ---------- | ---------- |
> | 160M       | L=3       | L=3        | L=4        | L=6        |
> | 470M       | L=3       | L=4        | L=4        | L=6        |
> | 1B         | L=3       | L=3        | L=5        | L=6        |
>
> [13] A Survey on Curriculum Learning. T-PAMI 2021.

---

### Official Review · Reviewer_WZPA · 2025-10-31

**Soundness:** 3
**Presentation:** 3
**Contribution:** 3
**Rating:** 6
**Confidence:** 4

**Summary:**

This paper proposes a new combination of data scoring and data ordering for LLM pretraining. Each datapoint is scored based on its learnability, i.e. how much it decreases loss during training, and its quality, i.e. how much the gradient of the loss w/r/t this datapoint aligns with the average gradient of the loss. For data ordering, the paper proposes "folding ordering", which takes the data scores and repeats curriculum learning several times rather than sorting data in ascending order by score. Comprehensive experiments show improvements over random shuffling and data selection baselines.

**Strengths:**

- The paper proposes a new combination of data scoring and data ordering methods and comprehensively evaluates them for LLM pretraining.
- The main results in Table 1 show that compared to randomly shuffled data, the proposed combination of data scoring and folding ordering yields improvements of up to 1.7 points of accuracy on 8 downstream tasks.
- The results showing that folding ordering outperforms random shuffling (Table 4) are very compelling, with folding outperforming shuffling in every evaluated setting, regardless of scoring method.

**Weaknesses:**

I have no major concerns about this paper. The primary weakness is that the improvements in data efficiency and efficacy are small beyond baselines. The results in Table 1 do show improvements over random shuffling, but it's not clear whether they warrant the added complexity of the approach.  For another example, in Table 3 where different scoring methods used with the same ordering method, the proposed method outperforms baselines for 4 out of 8 tasks, with an average improvement of 0.69 points of accuracy over the next best baseline.

Given the small improvements in general, the conclusions would be strengthened by replications in order to determine statistical significance, though I understand other papers on this topic typically do not do so because of the compute requirements. At the very least, including the standard deviation over the three random seeds for conventional random shuffling in Table 1 would help contextualize those numbers.

**Questions:**

- It seems like the learnability score in Equation 2 depends on the data ordering used for training. Do you account for this at all?
- The general form of the experimental setup in Section 5.1 should be described more clearly. My understanding is that 1) most experiments pre-train the Mistral architecture at various parameter sizes from scratch on subsets of the RedPajama dataset, and 2) Table 5 alone shows the effect of the method for post-training Qwen1.5-0.5B. What model and dataset size combinations are used for the pre-training results in Tables 2, 3, and 4?

Small comments:
- Figure 1 is so zoomed in that it exaggerates the differences between methods, in my opinion, by not showing the 0 point on the y-axis.
- Line 76: "Folding Learning (FO)" -> "Folding Ordering (FO)"
- Use `\citet{}` instead of `\citep{}` for in-text citations, like those in the first paragraph of Section 2.3.
- Line 219: "N is the quantity of \theta" -> I think you mean "N is the dimension of \theta"
- Line 380: "We presents" -> "We present"

---

> ### Author Response · Authors · 2025-11-24
> **Rebuttal 1: W1.1**
>
> Thank you for your insightful feedback. Your positive remarks about our work being "very compelling" and featuring "no major concerns" are highly motivating. Additionally, the weaknesses and questions you raised are insightful, and we have carefully addressed them below.
>
> > (W1.1) The primary weakness is that the improvements in data efficiency and efficacy are small beyond baselines. The results in Table 1 do show improvements over random shuffling, but it's not clear whether they warrant the added complexity of the approach.
>
> Thank you for raising this concern. We have provided a detailed explanation addressing the point on "small improvements". We hope this helps better appreciate the strengths of our work and the performance gains.
>
> 1. **The improvements are significant in condition of no increase in model and data scale.** It is note that data efficacy research aims to improve performance without incurring extra costs (e.g., increasing model parameters or data volume). Consequently, the numerical gains in this topic are relatively smaller compared to those achieved by scaling up model or data size. Our method achieves a maximum gain of 1.7% over random shuffling (Table 1). This improvement is substantial for this topic, exceeding the gains over random baselines reported by other advanced methods, such as data ordering (0.33% [1]) and data scoring methods like DSIR (1.18% [2]), and DiSF (1.70% [3]).
>
> 2. **Our method introduces limited computational overhead.** As detailed in Appendix E (LQS Computational Analysis), the computational cost of model training (**> 96h**) is significantly higher than that of sample scoring (**~12h, less than 13% of training time**), even when accounting for the entire scoring pipeline (proxy data annotation, scorer training, and inference). Additionally, it is important to clarify that the computational cost of LQS exhibits negligible growth as data or model sizes scale up. Furthermore, the overhead is amortized in practice:
>     - Reusable Scorer: Once trained, the data scorer can be directly applied to score other in-domain datasets (e.g., general language) very efficiently (**~15 mins for 10B tokens, less than 0.26% of training time**).
>     - Reusable Scored Data: The scored data can be reused to train models with different architectures or under various selection and ordering settings. This only requires performing data selection and ordering based on existing scores, which incurs negligible cost (**~2 mins for 10B tokens, less than 0.03% of training time**).
>
> [1] Strategic Data Ordering: Enhancing Large Language Model Performance through Curriculum Learning. Arxiv 2024.
>
> [2] Data selection for language models via importance resampling. Neurips 2023.
>
> [3] Combatting Dimensional Collapse in LLM Pre-training Data via Diversified File Selection. ICLR 2025.

---

> ### Author Response · Authors · 2025-11-24
> **Rebuttal 2: W1.2**
>
> > (W1.2) For another example, in Table 3 where different scoring methods used with the same ordering method, the proposed method outperforms baselines for 4 out of 8 tasks, with an average improvement of 0.69 points of accuracy over the next best baseline.
>
> Thank you for the question. We fully understand your concern regarding the limited improvements appearing in Table 3.
> We would like to clarify the specific purpose of Table 3 and provide additional results to clearly demonstrate the performance gap between our method and the baselines.
>
> 1. **Clarification on the purpose of Table 3.** In our paper, Table 3 is designed to demonstrate the effectiveness of our Folding Ordering (FO) strategy, rather than to evaluate the LQS scoring method. The results show that FO is universally effective across all data scoring metrics: Under the same data volume and without any selection, applying FO yields consistent gains over random shuffling for various scoring methods (gains: PDS +0.71%, DISR +0.91%, PDS +1.02%, LQS +1.71%). As discussed in our response to W1.1, these are substantial and stable improvements.
>
> 2. **A more comprehensive comparison (Table B1).** We apologize that the original presentation in Table 3 may have caused a misunderstanding regarding the full potential of our method. To address this, we present a comparison between our full method (LQS + FO) and other baselines (without FO) in Table B1. As shown in Table B1, our method significantly outperforms all baselines, achieving the following gains: +1.71% over Random, +1.70% over DSIR, and +1.27% over PDS.
>
> | Method | ARC-c | ARC-e | HS | LAMB | OBQA | PIQA | SciQ | Wino | Avg. |
> | :--- | :--- | :--- | :--- | :--- | :--- | :--- | :--- | :--- | :--- |
> | Random | 21.27 | 34.32 | 27.85 | 20.25 | 24.40 | 55.19 | 56.93 | 50.72 | 36.37 |
> | KenLM | 21.42 | 34.34 | 27.76 | 20.84 | 25.00 | 56.31 | 54.30 | 51.07 | 36.38 |
> | DSIR  | **22.69** | 34.55 | 28.26 | 21.81 | 24.40 | **56.80** | 54.80 | 51.14 | 36.81 |
> | PDS   | 21.84 | 35.02 | 27.61 | 19.93 | 24.80 | 56.23 | 59.00 | 51.38 | 37.01 |
> | LQS+FO (Ours) | 21.59 | **36.07** | **28.41** | **23.79** | **25.60** | 56.37 | **59.80** | **53.04** | **38.08** |
>
> Table B1. Comparison of our method with other data efficacy baselines. All reported baseline results reflect their highest scores across all selection ratios.

---

> ### Author Response · Authors · 2025-11-24
> **Rebuttal 3: W2, Q1**
>
> > (W2) Given the small improvements in general, the conclusions would be strengthened by replications in order to determine statistical significance, though I understand other papers on this topic typically do not do so because of the compute requirements. At the very least, including the standard deviation over the three random seeds for conventional random shuffling in Table 1 would help contextualize those numbers.
>
> Thank you for the suggestion regarding replications. It is helpful in enhancing the scientific rigor of our paper.
>
> First, we would like to emphasize that all "Conventional" (random shuffling) results reported in our paper have already represented the average over three random seeds. Additionally, following your suggestion, we add the standard deviation for the random shuffle baseline in Table 1 to provide a more objective context for our performance gains. And we also incorporate these updates into the revision (highlighted in ${\color{blue}\text{blue}}$).
>
> As shown in Table B2, the standard deviation of the random shuffle method across different seeds ranges **from 0.06% to 0.19%** on the final average (last column). Consequently, the fluctuation caused by random seeds is an order of magnitude smaller than the performance gain achieved by our method (**up to 1.71%**). This confirms that the improvements are driven by our method rather than random variance.
>
> Table B2. Results from the original Table 1 with standard deviations added. (Due to space constraints, we present a partial table here; please refer to the revision for the full version.)
>
> | Method | ARC-c | ARC-e | HS | LAMB | OBQA | PIQA | SciQ | Wino | Avg. |
> | :-| :-| :-| :-| :-| :-| :-| :-| :- | :-|
> | **Model size = 160M** | | | | | | | | | |
> | Conventional | $21.27_{±0.30}$ | $34.32_{±0.83}$ | $27.85_{±0.15}$ | $20.25_{±1.70}$ | $24.40_{±0.19}$ | $55.19_{±0.16}$ | $56.93_{±1.06}$ | $50.72_{±0.84}$ | $36.37_{±0.19}$ |
> | DELT (Ours) | **21.59** | **36.07** | **28.41** | **23.79** | **25.60** | **56.37** | **59.80** | **53.04** | **38.08** |
> | **Model size = 470M** | | | | | | | | | |
> | Conventional | $21.16_{±0.42}$ | $34.91_{±0.86}$ | $28.11_{±0.23}$ | $21.88_{±0.10}$ | $23.90_{±1.84}$ | $56.07_{±0.58}$ | $58.75_{±0.35}$ | $50.04_{±1.40}$ | $36.85_{±0.06}$ |
> | DELT (Ours) | **22.33** | **35.88** | **28.45** | **23.26** | **26.60** | **57.20** | **60.10** | **52.81** | **38.33** |
> | **Model size = 1B** | | | | | | | | | |
> | Conventional | $20.58_{±0.36}$ | $36.12_{±0.03}$ | $28.32_{±0.29}$ | $23.56_{±0.77}$ | $25.00_{±0.71}$ | $56.49_{±1.19}$ | $60.05_{±0.69}$ | **$52.07_{±0.17}$** | $37.77_{±0.07}$ |
> | DELT (Ours) | **22.76** | **37.95** | **29.95** | **26.38** | **26.00** | **58.07** | **60.90** | 51.28 | **39.17** |
>
> > (Q1) It seems like the learnability score in Equation 2 depends on the data ordering used for training. Do you account for this at all?
>
> Thank you for the question. We would like to clarify that the order of samples within $D_{prx}$ does not affect the scoring results.
>
> 1. **Clarification on the LQS implementation process**. To address your concern, we would like to clarify the exact procedure for calculate the scores.
>     - Step 1: We first prepare a pre-trained 160M Mistral model $\theta_0$ (loaded from [4]). Then, we randomly sample two non-overlapping datasets, $D_{aux}$ and $D_{prx}$, from the training data $D$.
>     - Step 2: We initialize the proxy LM with $\theta_0$ and continue training on $D_{aux}$ for $T$ steps, saving the checkpoint at every step. Let the set of checkpoints be denoted as $\Theta = [\theta_0, \theta_1, ..., \theta_T]$.
>     - Step 3: We use these checkpoints to calculate the score for each corresponding sample ${x}\_n$ in the proxy dataset $D_{prx}$. The score $\gamma\_n$ includes the learnability score (Equation 2) you mentioned and is calculated as: $ \gamma\_n = \sum_{t=1}^{T-1} \frac{{\lambda}\_{t+1}^{\top} \nabla \ell({x}\_n, {\theta}\_t)}{\| \nabla \ell({x}\_n, {\theta}\_{t+1}) \|} $
>     - **Conclusion.** In this process, the set of checkpoints $\Theta$ is derived from the training trajectory on $D\_{aux}$. Crucially, during the scoring phase, each sample $x_n$ belongs to the proxy dataset $D\_{prx}$, which is a set completely distinct from $D\_{aux}$. Furthermore, the score for every ${x}\_n$ is computed by aggregating the gradients across all checkpoints in $\Theta$ (via the summation $\sum_{t=1}^{T-1}$). Therefore, the calculation of our LQS score (including the learnability score) is independent of the order of the samples ${x}\_n$ in $D\_{prx}$  being scored.
>
> 2. **Revision to improve clarity.** We apologize if the original description in the method section caused confusion. We have moved the detailed implementation steps described (originally in Appendix) to the main text to ensure transparency and understandable. Please refer to ${\color{blue}\text{Section 3.3}}$ in the revision, highlighted in ${\color{blue}\text{blue}}$.
>
> [4] https://huggingface.co/Data-Selection/PDS-160M

---

> ### Author Response · Authors · 2025-11-24
> **Rebuttal 4: Q2, Q3**
>
> > (Q2) The general form of the experimental setup in Section 5.1 should be described more clearly. My understanding is that 1) most experiments pre-train the Mistral architecture at various parameter sizes from scratch on subsets of the RedPajama dataset, and 2) Table 5 alone shows the effect of the method for post-training Qwen1.5-0.5B. What model and dataset size combinations are used for the pre-training results in Tables 2, 3, and 4?
>
> We apologize for the confusion caused by our ambiguous wording. We have clarified the experimental setup and added a detailed table to the revision to ensure transparency.
>
> First, for Tables 2, 3, and 4, we employed the **160M Mistral** model and conducted all experiments on the **same 1B-token** dataset, which was randomly sampled from RedPajama [5].
>
> Second, while these model and data details were originally mentioned in Appendix B (Additional Experimental Setup), we have now presented them in a tabular format to provide a more precise overview of the training configuration. As shown in Table B3, these updates have been included in our revision (highlighted in ${\color{blue}\text{blue}}$ in Appendix B).
>
> Table B3. Detailed experimental setting.
>
> | Category | Table 1 (a) | Table 1 (b) | Table 1 (c) | Table 2 | Table 3 | Table 4 | Table 5 (a) | Table 5 (b) |
> | :--- | :--- | :--- | :--- | :--- |:--- | :--- | :--- | :--- |
> | Goal | Robustness across different model sizes (Pre-train) | Robustness across different data scales (Pre-train) | Scalability to relatively large model size and data scale (Pre-train) | Comparison of different ordering methods using the same scoring method (LQS) (Pre-train) | Comparison of different scoring methods using the same ordering method (FO) (Pre-train) | Robustness across different data selection ratios (Pre-train) | Robustness on the Math domain (Post-train) | Robustness on the Code domain (Post-train) |
> | Data | 1B-token data from RedPajama | 10B-token, 50B-token data from RedPajama | 50B-token data from RedPajama | 1B-token data from RedPajama | 1B-token data from RedPajama | Subsets of the 1B-token RedPajama data based on selection ratios from 0.5 to 1.0. | 1B-token data from OpenWebMath | 1B-token data from The Stack v2 |
> | Model | 160M, 470M, 1B Mistral | 160M Mistral | 1.7B Mistral | 160M Mistral | 160M Mistral | 160M Mistral | Qwen1.5-0.5B, 1.8B | Qwen1.5-0.5B, 1.8B |
>
> [5] RedPajama: an Open Dataset for Training Large Language Models. NeurIPS 2024.
>
> > (Q3) Small comments:
> Figure 1 is so zoomed in that it exaggerates the differences between methods, in my opinion, by not showing the 0 point on the y-axis.
> Line 76: "Folding Learning (FO)" -> "Folding Ordering (FO)"
> Use instead of for in-text citations, like those in the first paragraph of Section 2.3.\citet{}\citep{}
> Line 219: "N is the quantity of \theta" -> I think you mean "N is the dimension of \theta"
> Line 380: "We presents" -> "We present"
>
> Thank you for your careful proof reading and valuable suggestion regarding our paper. We have implemented the following changes in our revision, highlited in ${\color{blue}\text{blue}}$.
>
> 1. First, we revise Figure 1 in the revision to present a more objective comparison in ${\color{blue}\text{Appendix F.1 Revised Figure 1 with adjusted y-axis}}$.
> Second, we would like to provide further context regarding the visualization choice. As discussed in our response to W1.1, our method achieves a performance gain of up to 1.71% over baselines in Figure 1, this is considered significant in the context of data-efficient pre-training (which entails no extra compuations or parameters). Given these fine-grained but impactful improvements, it is a standard practice in this domain to adjust the y-axis scale to visualize the differences clearly. This convention is also adopted by other leading works such as PDS [6] (Figures 1, 4–7), DSIR [7] (Figure 3), and DiSF [8] (Figures 4–8).
>
> 2. We correct "Folding Learning" to "**Folding Ordering**" to ensure the consistency and accuracy of the abbreviation.
>
> 3. We change the citations in the first paragraph of Section 2.3 to in-text citations (using `\citet{}`). We have also checked the entire manuscript to ensure consistent usage and avoid mixing `\citet{}` and `\citep{}`.
>
> 4. We revise the statement to a more rigorous and precise formulation: "**$N$ is the dimension of the parameter space.**"
>
> 5. We have corrected "We presents" to "**We present**" in Line 363 (Former Line 380). We have also checked the entire manuscript to ensure there are no similar grammatical errors.
>
> Your valuable suggestions greatly contribute to improving the quality of our paper. If you have any questions, please don't hesitate to ask. Thank you again for your time and expertise.
>
> [6] Data Selection via Optimal Control for Language Models. ICLR 2025.
>
> [7] Data selection for language models via importance resampling. NeurIPS 2023.
>
> [8] Combatting Dimensional Collapse in LLM Pre-training Data via Diversified File Selection. ICLR 2025.

---

### Official Review · Reviewer_xC4U · 2025-11-01

**Soundness:** 2
**Presentation:** 2
**Contribution:** 2
**Rating:** 4
**Confidence:** 3

**Summary:**

The paper studies data efficacy for language model training with aim at improving model performance by optimizing the way of utilizing data. It features two strategies for data scoring and ordering, called folding ordering and learnability-quality scoring, respectively. Optionally, data selection, for efficiency, can be incorporated between the applications of two strategies in the proposed pipeline called DELT. Extensive experiments are conducted to demonstrate the superiority of this strategy combination.

**Strengths:**

1. The two strategies proposed for improving data utilization looks simple yet effective
2. Propose a general framework for training data utilization
3. Extensive experiments are conducted to verify the effectiveness of the proposed strategies for utilizing data

**Weaknesses:**

1. It remains unknown at all levels except for experiments why folding ordering is better than curriculum learning.
2. The data scoring strategy lacks theoretical support, and is more like heuristics
3. LQS is calculated only using a small number of data points, which raises news questions on how to select these data points and how to ensure no bias
4. The experimental setting could be made clearer, e.g., by drawing the whole workflow that covers all about the setting
5. In Line 432, "For the LQS method, it outperforms other baselines across different ordering methods and selection ratios". This claim seems incorrect. For example, LQS performs worst for folding ordering  and selection ratio .9 in Table 4

**Questions:**

1. In Line 190-191, "Folding learning not only inherits the benefits from curriculum learning but also mitigates issues of model forgetting, and data distribution bias". Why?
2. In Line 232 "Conversely, noisy samples or those with unstable gradients yield a low learnability score". Why?
3. In Line 244, the target vector $\lambda$ is calculated by following Gu et al. (2025). What's the connection of your strategies to this work and why?
4. For data selection, the score vector is sorted in ascending order and the last K samples are discarded. But    it is mentioned in Line 254 that "samples with large score values have higher quality and significant contributions to reducing the downstream loss J(θ), especially when introduced during later training stages". Does this mean that the top-K samples are discarded?
5. In Section 5, what does "conventional" mean exactly in Table 1? It hides a lot of details. What's the metric for results there?

---

> ### Author Response · Authors · 2025-11-24
> **Rebuttal 1: W1, Q1**
>
> Thank you for your thoughtful comments. We deeply appreciate your recognition of our work as "simple yet effective", and "extensive experiments" which greatly encourages us.
>
> You also raised some points of weakness and questions, which we found very enlightening. We thoroughly analyzed the issues you mentioned and provided the following responses.
>
> > (W1) It remains unknown at all levels except for experiments why folding ordering is better than curriculum learning.
>
> > (Q1) In Line 190-191, "Folding learning not only inherits the benefits from curriculum learning but also mitigates issues of model forgetting, and data distribution bias". Why?
>
> Thank you for your penetrating question. We fully understand your concerns regarding the empirical results of folding ordering. We apologize for not sufficiently explaining the confusion caused by model forgetting and data distribution bias associated with Curriculum Learning (CL) in the paper.
>
> In traditional CL, previous works [2,5] have already pointed out that model forgetting and data distribution bias are two widely recognized issues, which become severe in LLM training where only one or few epochs are applied.
>
> 1. **Model forgetting.** Traditional CL starts training from the easiest data bucket [1]. This sequential curriculum causes the model to lose contact with previously learned data, leading to the catastrophic forgetting of even simple concepts learned early on [2]. **Existing Solutions** address model forgetting by using auxiliary models to detect 'forgetting' signals [3,4], dynamically increasing the replaying probability of data soon to be forgotten [9]. However, utilizing auxiliary models introduces substantial computational overhead. Additionally, simple replay strategies are incompatible with the single-epoch setting of mainstream LM pre-training, making these approaches impractical.
> 2. **Optimization traps due to data distribution bias.** Traditional CL restricts the model to only "easy sample subsets" during early training. This high homogeneity may restrict the gradient descent optimization path, causing the model to converge to a difficult-to-escape sharp local minimum [5]. **Existing Solutions** often involve regularization or mixing strategies to enforce the selection of more diverse samples, thereby avoiding local optima specific to a single difficulty group [6,7]. However, these methods often require designing specialized diversity metrics tailored to specific datasets, rendering them impractical.
>
> To mitigate these recognized issues, we introduced FO, a simple yet effective strategy. Specifically, it degenerates to traditional CL at $L=1$ and becomes random shuffling at $L \to \infty$. When configured with an appropriate value, FO effectively balances the sequential learning capability of CL with the robustness of random shuffling against forgetting and distribution bias.
>
> We validate its superior performance over traditional CL in addressing these two problems through the following experimental evidence.
>
> 3. **Validation on model forgetting.** The sequential nature of CL (easy $\to$ hard) can lead to the irreversible overwriting of "simple" knowledge. By introducing $L$ folds, the model effectively simulates the process of repeating earlier knowledge (pseudo-replay), thereby preventing knowledge forgetting.
>
>     To directly monitor the forgetting phenomenon, we designed a layered validation experiment. We track the Perplexity (PPL) on the easiest 10% of the validation set ($D_{easy}$) throughout training. As shown in ${\color{blue}\text{Appendix F.2.2}}$) of the revision, the PPL of traditional CL (sorting) on $D_{easy}$ rapidly drops (initial 30% phase) but then significantly rebounds after entering the "hard" data region (latter 50%), which directly confirms the forgetting of simple samples. For the FO (L=3), PPL drops normally in Fold 1. When simple data is reintroduced in Fold 2, PPL shows a secondary sharp drop (Re-learning). By the end of training (Fold 3), the model maintains an extremely low PPL on $D_{easy}$ and exhibits no rebound phenomenon seen in CL.
> 4. **Validation on optimization traps.** The strict ordering in CL leads to high difficulty homogeneity within training batches. Theoretically, FO resolves this by folding, which intentionally reintroduces samples of differing difficulty within each micro-training window, restoring local data diversity and acting as an implicit regularizer.
>
>     We quantitatively analyzed the training data flow's difficulty diversity (measured by the standard deviation of sample scores within a sliding window, see ${\color{blue}\text{Appendix F.2.1}}$). The results show that traditional CL shows low difficulty diversity, confirming its severe distribution bias. In contrast, FO boosts this diversity (an increase of ~4.7x). This indicates that FO mitigates the local data distribution bias inherent in CL, helping the optimizer converge on a smoother and more representative loss landscape.

---

> ### Author Response · Authors · 2025-11-24
> **Rebuttal 2: W2, W3**
>
> > (W2) The data scoring strategy lacks theoretical support, and is more like heuristics.
>
> We thank the reviewer for raising this insightful question. We fully understand your concern regarding theoretical support, as a lack of rigorous mathematical grounding is indeed a common issue in many existing data selection works for language model pre-training, which often rely heavily on heuristics (e.g., perplexity filtering or semantic matching) [10-13].
>
> However, we wish to emphasize that our method is NOT a heuristic design. Instead, it is built upon an established theoretical foundation designed specifically for data scoring in PDS [14], and we have further extended this theory to address the stability challenges in practical deep learning optimization. Specifically, our Learnability-Quality Scoring (LQS) strategy can be theoretically characterized as classical Optimal Control theory [15].
>
> 1. LQS is rooted in the PDS theoretical framework. Our approach directly inherits the theoretical foundation of PDS [14]. PDS is the first work to formally formulate LM pre-training as an Optimal Control Problem, deriving the necessary conditions for optimal data selection via Pontryagin's Maximum Principle (PMP).
>     - Theoretical Correspondence: The numerator of our LQS formula, which is specifically represented by $\lambda_{t+1}^\top \nabla l(x_n, \theta_t)$, directly corresponds to the Optimality Condition derived in PDS (Theorem 2.1, Eq. 6).
>     - The target vector $\lambda$ in our work is also not a heuristic definition but is mathematically equivalent to the Co-state Vector in optimal control theory. It encodes global gradient information back-propagated from future timesteps via the Hessian matrix, representing the sensitivity of the final objective to the current state.
>     - This confirms that the core driver of LQS is derived from rigorous mathematical principles rather than heuristics.
>
> 2. LQS Improves PDS via "Stability Regularization". While PDS provides the optimal solution under ideal control assumptions, applying raw PMP conditions directly to the non-convex and stochastic optimization landscape of deep neural networks presents challenges. Raw PMP tends to favor samples with the largest gradient magnitudes (maximizing the inner product). In practice, such "large gradient" samples may contain outliers or high noise, leading to training instability. Our innovation lies in introducing the denominator $||\nabla l(x_n, \theta_{t+1})||$ as a "Stability Regularizer":
>     - Optimization perspective: This term measures the local convergence speed and stability of a sample (Learnability). If a sample's gradient norm decreases significantly after an update (small denominator), it indicates the information is effectively "absorbed" by the model. Conversely, a persistently large gradient norm implies high uncertainty or an outlier.
>     - Theoretical implication: By normalizing the PMP objective with the future gradient norm, LQS effectively maximizes the alignment with the optimal direction (Quality) while imposing a constraint on local optimization stability. As shown in our experiments (Figure 1), this Regularized Optimal Control strategy outperforms the pure PDS baseline, demonstrating the necessity of this theoretical refinement for practical efficacy.
>
> 3. Clarification on theoretical exposition. Given that PDS [14] has already provided a comprehensive derivation of the underlying Optimal Control theory and PMP proofs, our manuscript focused primarily on the new paradigm of Data Efficacy and the practical introduction of "Learnability" into this theoretical framework. Consequently, we avoid restating the raw PMP theory in our work.
>
> In summary, LQS can be physically interpreted as a stability-regularized solution to the optimal control problem. We hope this clarification addresses your concerns regarding the theoretical grounding of our scoring strategy.
>
> > (W3) LQS is calculated only using a small number of data points, which raises news questions on how to select these data points and how to ensure no bias.
>
> Thank you for this insightful question. As the proxy data ${D}_{prx}$ used for LQS annotation is a subset of the complete dataset ${D}$, the challenge of entirely avoiding sampling bias is a valid concern.
>
> In our current implementation, we mitigate this by using **random sampling** to select ${D}_{prx}$ from ${D}$. Random selection is universally accepted as the most practical and effective method for maximizing the preservation the underlying distribution of ${D}$.

---

> ### Author Response · Authors · 2025-11-24
> **Rebuttal 3: W4, W5**
>
> > (W4) The experimental setting could be made clearer, e.g., by drawing the whole workflow that covers all about the setting.
>
> We apologize for the confusion caused by our ambiguous wording. We have clarified the experimental setup and added a detailed table to the revision to ensure transparency and provide a more precise overview of the training configuration. As shown in Table A3, these updates have been included in our revision (highlighted in ${\color{blue}\text{blue}}$ in Appendix B).
>
> Table A1. Detailed experimental setting.
>
> | Category | Table 1 (a) | Table 1 (b) | Table 1 (c) | Table 2 | Table 3 | Table 4 | Table 5 (a) | Table 5 (b) |
> | :--- | :--- | :--- | :--- | :--- |:--- | :--- | :--- | :--- |
> | **Goal** | Robustness across different model sizes (Pre-train) | Robustness across different data scales (Pre-train) | Scalability to relatively large model size and data scale (Pre-train) | Comparison of different ordering methods using the same scoring method (LQS) (Pre-train) | Comparison of different scoring methods using the same ordering method (FO) (Pre-train) | Robustness across different data selection ratios (Pre-train) | Robustness on the Math domain (Post-train) | Robustness on the Code domain (Post-train) |
> | **Data** | 1B-token data from RedPajama | 10B-token, 50B-token data from RedPajama | 50B-token data from RedPajama | 1B-token data from RedPajama | 1B-token data from RedPajama | Subsets of the 1B-token RedPajama data based on selection ratios from 0.5 to 1.0. | 1B-token data from OpenWebMath | 1B-token data from The Stack v2 |
> | **Model** | 160M, 470M, 1B Mistral | 160M Mistral | 1.7B Mistral | 160M Mistral | 160M Mistral | 160M Mistral | Qwen1.5-0.5B, 1.8B | Qwen1.5-0.5B, 1.8B |
>
> > (W5) In Line 432, "For the LQS method, it outperforms other baselines across different ordering methods and selection ratios". This claim seems incorrect. For example, LQS performs worst for folding ordering and selection ratio .9 in Table 4.
>
> Thank you for your careful review, which has greatly helped improve the precision of our claims. We have revised our statement as follows (highlighted in ${\color{blue}\text{blue}}$ in our revision at  ${\color{blue}\text{Line 123}}$):
>
> "For the LQS method, it outperforms other baselines under the random shuffle setting across most selection ratios."
>
> We have also carefully checked the entire manuscript to avoid similar issues.

---

> ### Author Response · Authors · 2025-11-24
> **Rebuttal 4: Q2**
>
> > (Q2) In Line 232 "Conversely, noisy samples or those with unstable gradients yield a low learnability score". Why?
>
> Thank you for this insightful question regarding noisy samples. The Learnability score is one of the key novelty of our work, and we provide a detailed explanation from both theoretical and experimental perspectives to clarify why noisy samples yield lower scores.
>
> **Theoretical justification.** Recall the definition of the Learnability Score for a sample $x$: $\mathcal{L}(x) = \sum_{t=1}^{T-1} \frac{\|
> \nabla l(x, \theta_t)\|}{\|
> \nabla l(x, \theta_{t+1})\|}$. Let $g_t = \|
> \nabla l(x, \theta_t)\|$ denote the gradient norm at step $t$. Below, we categorize the data into high-quality and noisy subsets, then provide a theoretical derivation demonstrating that $\mathcal{L(x_{High-Quality})} > \mathcal{L(x_{Noisy})}$ holds under realistic optimization conditions.
>
> Case A: High-Quality / Learnable Samples. These samples provide consistent training signals, leading to a stable reduction in gradient norm as the model learns. While local fluctuations exist, the dominant trend is a structural reduction in gradient error as the model converges. We model this as a process with a consistent contraction rate $\alpha > 0$ and a minor fluctuation term $\xi_t$: $g_{t+1} = (1 - \alpha)g_t + \xi_t$.
>
> Since the "structural descent" dominates the fluctuation (as per the condition that the overall downward trend is significant), we have $g_{t+1} < g_t$ on average. The ratio $r_t \approx \frac{1}{1-\alpha} \approx 1 + \alpha$. Over $T$ steps, the score accumulates this positive learning signal: $\mathcal{L}(x_{A}) \approx \sum_{t=1}^{T} (1 + \alpha) = T + T\alpha$. Here, $T\alpha$ represents the accumulated reward from continuously reducing the gradient norm.
>
> Case B: Noisy samples represent aleatoric uncertainty (e.g., random labels or incoherent text). The model cannot systematically minimize the loss, the gradient norm does not decay towards zero but fluctuates around a noise floor. We model this as a random walk or stagnation around a constant level: $g_{t+1} = g_t + \epsilon_t$,
> where $\epsilon_t$ is a random noise term. Unlike Case A, there is no consistent contraction driver $\alpha$.
>
> Even if Case B experiences occasional large downward fluctuations (yielding high single-step rewards), the consistency and total magnitude of Case A's descent far exceed the random fluctuations of Case B. Specifically, the accumulated learning signal of Case A ($\sum \alpha g_t$) is orders of magnitude larger than the net drift of Case B ($\sum \epsilon_t$). Since the fluctuations $\epsilon_t$ are random and do not compound into a structural descent, they largely cancel out or result in a negligible net change over $T$: $\mathcal{L}(x\_{B}) \approx \sum\_{t=1}^{T} 1 = T$
>
> **Conclusion.** Comparing the two cumulative scores:$\mathcal{L}(x_{A}) \approx T + T\alpha \gg \mathcal{L}(x_{B}) \approx T$. The inequality $\mathcal{L}(x_{A}) > \mathcal{L}(x_{B})$ holds because the systematic learning term $T\alpha$, which accumulated from consistent gradient reduction, strictly dominates the baseline score $T$ obtained by noisy samples. Even if a noisy sample luckily obtains a high ratio in a single step $t$, it lacks the temporal consistency ($\alpha$) to sustain this reward over the full trajectory, ensuring it receives a lower total score than a learnable sample.
>
> **Experimental verification.** To empirically validate this, we added the visualization of the gradient norm dynamics in ${\color{blue}\text{Appendix G.1}}$. As shown in Figure 10, the learnability score from high-quality samples (gradient reduction) consistently outweighs the variance from noisy samples, ensuring that learnable high-quality data is assigned significantly higher scores. We have updated the revision (${\color{blue}\text{Appendix G.1}}$, highlighted in ${\color{blue}\text{blue}}$) with these visualizations and analyses to further clarify this mechanism.

---

> ### Author Response · Authors · 2025-11-24
> **Rebuttal 5: Q3, Q4**
>
> > (Q3) In Line 244, the target vector is calculated by following Gu et al. (2025). What's the connection of your strategies to this work and why?
>
> Thank you for your question. We elaborate on the connections and distinctions between our work and PDS [14] (Gu et al., 2025), which we hope provides a clear understanding of their relationship.
>
> The connections are as follows.
>
> 1. **Theoretical connection: the foundation of the target vector.** As mentioned in our response to W2, our method leverages the strong theoretical foundation established by PDS, which was the first to formulate data selection for LMs pre-training as an Optimal Control problem. The definition of $\lambda$ (target vector) in our work is mathematically equivalent to the Co-state Vector derived in PDS using Pontryagin's Maximum Principle (PMP).
>
> 2. **The reason we use it.** Unlike heuristic targets (e.g., average gradients), this co-state vector $\lambda$ uniquely encodes the global training dynamics. It captures how a parameter update at the current step impacts the final downstream loss by back-propagating gradient information from future timesteps (via the Hessian). By adopting this vector, our Quality metric inherits the theoretical guarantee of pointing towards the global optimum of the control trajectory.
>
> While we adopt the target vector definition from PDS, our strategies diverge significantly in motivation, scoring, and ordering methods.
>
> 1. **Motivation and task.** PDS focuses on data selection (efficiency), which identifies a sparse subset to reduce training cost. In contrast, our work introduces the broader concept of data efficacy. We focus on maximizing the performance gain from a fixed or selected budget by optimizing data utilization (i.e., how and when to present data).
>
> 2. **Scoring strategy.** PDS scores data purely based on the inner product $\lambda^\top \nabla l$ (Quality). Ideally, this maximizes the Hamiltonian function under PMP. However, in non-convex optimization, this can favor samples with exploding gradients or high variance. Therefore, we propose Learnability-Quality Scoring (LQS), which divides the PDS quality term by a learnability term (the future gradient norm $||\nabla l_{next}||$). This acts as a "Stability Regularizer" applied to the PDS theory. It penalizes samples that are directionally correct but difficult to converge (high uncertainty), ensuring we prioritize data that is both high-quality and efficiently learnable by the current model state. This allows LQS to serve both ordering and selection tasks more robustly than the raw PDS score.
>
> 3. **Ordering strategy.** Since PDS is designed for selection, it does not address the order of data presentation. We propose Folding Ordering, a novel scheduling strategy based on our LQS. FO leverages the "Learnability" component of our score to construct a learning process that mitigates the distribution bias and forgetting issues common in traditional sorting-based curricula.
>
> In summary, while PDS provides the "compass" (target vector $\lambda$), our work designs a better "path" (LQS and Folding Ordering) to reach the destination efficiently and stably.
>
> > (Q4) For data selection, the score vector is sorted in ascending order and the last K samples are discarded. But it is mentioned in Line 254 that "samples with large score values have higher quality and significant contributions to reducing the downstream loss J(θ), especially when introduced during later training stages". Does this mean that the top-K samples are discarded?
>
> Yes, you are absolutely correct about the data selection. Our method assigns higher scores to better samples (indicating high quality and learnability). This is a common practice in data selection for both language models [13,14,16-18] and vision models [19-21].
>
> Therefore, when performing data selection on our scored dataset $D$, we sort the samples in ascending order. We then select the last $K$ samples (the highest-scoring ones) based on a preset selection ratio $r$, and discard the first $|D| - K$ samples, where $K = |D| \times r$.
>
> Thank you for pointing this out. We realize that the "Top-k" label in the Data Selection module of **Figure 6** (former Figure 4) may have caused confusion. We have corrected it to Bottom-K at Figure 6 in the revised version.

---

> ### Author Response · Authors · 2025-11-24
> **Rebuttal 6: Q5**
>
> > (Q5.1) In Section 5, what does "conventional" mean exactly in Table 1?
>
> We apologize for the confusion caused by our ambiguous wording.
>
> In contrast to our method (which involves specific scoring, selection, and ordering), the "conventional" refers to training on randomly shuffled data without any selection.
> Additionally, to ensure reliability, the conventional method reports the average result over three random seeds (as noted in the caption of Table 1).
>
> To address this issue, we have also added this clarification in our revision ${\color{blue}\text{Section 5.1 (Baselines, L356)}}$: "Conventional (randomly shuffled data order without selection)."
>
> > (Q5.2) It hides a lot of details. What's the metric for results there?
>
> Thank you for the question. The benchmarks in Table 1 are formulated as multiple-choice tasks, where the model selects the answer by minimizing the normalized loss across candidate options (`acc_norm`).
>
> All evaluations were implemented using the Language Model Evaluation Harness (https://github.com/EleutherAI/lm-evaluation-harness). Full details regarding the evaluation setup, including specific benchmarks and metrics, are provided in **Appendix B Additional Experimental Setup**.
>
> We hope this response helps clarifie the details of our work. We are grateful for your professional and valuable suggestions, which have significantly enhanced the quality and rigor of our paper. Please don't hesitate to ask if you have any further questions.
>
> [1] Curriculum learning. ICML 2009.
>
> [2] A Survey on Curriculum Learning. T-PAMI 2021.
>
> [3] Teacher-student curriculum learning. TNNLS 2019.
>
> [4] Automated curriculum learning for neural networks. ICML 2017.
>
> [5] Curriculum Learning: A Survey. IJCV 2022.
>
> [6] Self-paced learning with diversity. Neurips 2014.
>
> [7] Minimax curriculum learning: Machine teaching with desirable difficulties and scheduled diversity. ICLR 2018.
>
> [8] LAMOL: LAnguage MOdeling for Lifelong Language Learning. ICLR 2020.
>
> [9] Experience replay for continual learning. Neurips 2019.
>
> [10] Semdedup: Data-efficient learning at web-scale through semantic deduplication. ICLR 2023.
>
> [11] D4: Improving llm pretraining via document de-duplication and diversification. Neurips 2023.
>
> [12] When Less is More: Investigating Data Pruning for Pretraining LLMs at Scale. Neurips 2023.
>
> [13] Data Selection for Language Models via Importance Resampling. Neurips 2023.
>
> [14] Data Selection via Optimal Control for Language Models. ICLR 2025.
>
> [15] Optimal control. John Wiley & Sons, 2012.
>
> [16] QuRating: Selecting High-Quality Data for Training Language Models. ICML 2024.
>
> [17] The FineWeb Datasets: Decanting the Web for the Finest Text Data at Scale. Neurips 2024.
>
> [18] Meta-rater: A Multi-dimensional Data Selection Method for Pre-training Language Models. ACL 2025.
>
> [19] Deep Learning on a Data Diet: Finding Important Examples Early in Training. Neurips 2021.
>
> [20] Coverage-centric Coreset Selection for High Pruning Rates. ICLR 2023.
>
> [21] Training-Free Dataset Pruning for Instance Segmentation. ICLR 2025.

---

### Author Response · Authors · 2025-12-01
**General Response**

Dear AC, SAC, and PC,

We sincerely appreciate the time and effort you and the reviewers have dedicated to our paper.

Firstly, **it is a great honor to have the AC overseeing our submission**. We deeply appreciate the AC's efforts in managing the review process and enhancing its quality, which greatly benefits the entire academic community.

Secondly, the author team would like to once again highlight the **key contributions and novelty** of this paper.

- We are the first to explicitly define **data efficacy** for language model training, a concept that is distinct from data efficiency. While data efficiency focuses on reducing dataset size (e.g., data selection), data efficacy aims to improve model performance through optimizing the utilization of training data (e.g., data ordering). Notably, data efficacy and efficiency can be applied simultaneously in LM training. We hope this work paves the way for establishing data efficacy as a new track.

- To enhance data efficacy, we propose a novel data ordering method, termed **Folding Ordering**. Unlike previous Curriculum-learning-based methods, which directly sort data by difficulty and often lead to model forgetting and data distribution bias, our approach effectively mitigates these challenges.

- Furthermore, to facilitate effective data ordering, we introduce a novel **learnability-quality scoring** method. Unlike previous scoring methods designed solely for data selection, our learnability-quality score combines multiple aspects to determine a final score that effectively serves the dual purpose of data ordering and data selection.

- Extensive experiments show that our method enhances pre- and post-training performance across various model sizes and data scales, offering an efficient boost while being easily pluggable into existing data curation pipelines.

Finally, we also thank the reviewers for their insightful comments. We are encouraged that they recognized our work establishes an 'impactful foundation,' describing the method as 'innovative' and 'simple yet effective,' and the experiments as 'thorough,' 'persuasive,' and 'very compelling.

The constructive feedback provided new perspectives that have significantly contributed to improving the paper's quality. In response to the reviews, we provide detailed clarifications, additional theoretical analysis, and new experiments. All revised sections in the manuscript are highlighted in ${\color{blue}\text{blue}}$ for easy reference.

Our key updates include:

1. **Additional Analysis & Experiments.**

    - **Analysis on model forgetting and optimization traps.** We provide a theoretical and empirical analysis of the issues caused by traditional Curriculum Learning (CL), specifically model forgetting and optimization traps due to data bias. We align our explanations with widely recognized principles and validated our Folding Ordering (FO) method's effectiveness through new experiments. These include contrasting PPL dynamics on simple samples during training (those appearing early in the CL schedule) and analyzing local difficulty diversity, both of which confirm that FO significantly mitigates these issues (Appendix F.2.2, F.2.3).

    - **Analysis on the Learnability Score.** As a core innovation of LQS, we provide a theoretical proof demonstrating how the Learnability Score distinguishes between noisy and hard samples. We further validate this design via gradient norm visualization experiments, confirming that the scoring functions exactly as intended (Appendix G.1).

2. **Clarifications.**
    - **Theoretical foundations of LQS.** We explain the solid theoretical basis of our LQS method, detailing how it builds upon and advances established theories.

    - **Scalability analysis.** Leveraging our empirical results (160M–1.7B models; 1B–50B tokens) and the Chinchilla Scaling Law, we extrapolate our performance to larger scales. The results demonstrate that the improvements yielded by our method (LQS & FO) persist in pre-training large LMs, such as GPT-3 (175B) and the Llama family (6.7B–405B).

    - **Significance of performance gains by our method.** We clarify that our method achieves substantial gains compared to other methods in this topic, particularly given that these improvements are achieved without increasing data scale or model size.

    - **Manuscript polishing.** We have thoroughly proofread the manuscript to correct typographical errors and improve overall readability.

Following our detailed responses, Reviewer dGm4 (Rating 2) has indicated an intention to raise the score and other three reviewers (Ratings 6, 6, 4) have not yet responded. We believe the revisions and clarifications outlined above effectively address the reviewers’ concerns and strengthen the paper. We hope these changes clarify our contributions and the robustness of our findings, and we would be grateful for the committee’s favorable consideration.

Best regards,

All authors of Submission 2043

---

### Meta-Review · Area_Chair_kZEi · 2025-12-24

**Summary:**

This paper studies optimizing the utilization of training data via joint data selection and ordering within training. Numerically, the framework shows modest gains over baselines.  Multiple reviewers expressed concern with the organization of the paper, including motivation and novelty of the proposed function. A second core concern is the limited performance improvement and questions about whether they will scale to larger models. The rebuttal includes a revised draft that attempts to clarify some of these points. However, (as one reviewer notes), the key motivation justifying the framework is still unclear. This, in addition to the limited statistical analysis demonstrating the significance of the gains suggest that the paper can be improved for a resubmission in a later conference cycle.

**Reviewer Concerns:**

- Weak motivation for folding learning and unclear writing: The rebuttal provides some suggestions to motivate the framework. The revised paper includes more clarification.
- Limited novelty for proposed quality and reliability functions: The rebuttal does not fully clarify how the two components of folding learning nd LQS scoring improve over curriculum learning or PDS.
- Limited validation on larger models: The rebuttal uses Chinchilla to fit scaling laws and forecast performance at large scale. This is reasonable, given the compute cost of larger training. However, the gains appear minimal.
- Improvements are small and necessity for statistical significance: The rebuttal argues that the gains are meaningful with respect to the additional compute cost. The rebuttal includes some repetition over multiple seeds for random shuffling, but not for all models.

**Reviewer Scores:**

Of the two negative reviewers, one reviewer stated that they slightly raised their score to a weakly negative review but they could not recommend acceptance. Given that the remaining negative reviewer shared similar concerns about unclear support for the proposed approach, I do not believe they would raise their score to acceptance.

---

### Decision · Program_Chairs · 2026-01-26

Reject